# Algorithms and Hardness for Learning Linear Thresholds from Label Proportions

**Rishi Saket**
Google Research India
rishisaket@google.com

## Abstract

We study the learnability of linear threshold functions (LTFs) in the learning from label proportions (LLP) framework. In this, the feature-vector classifier is learnt from bags of feature-vectors and their corresponding observed label proportions which are satisfied by (i.e., consistent with) some unknown LTF. This problem has been investigated in recent work ([37]) which gave an algorithm to produce an LTF that satisfies at least $(2/5)$-fraction of a satisfiable collection of bags, each of size $\leq 2$, by solving and rounding a natural SDP relaxation. However, this SDP relaxation is specific to at most 2-sized bags and does not apply to bags of larger size.

In this work we provide a fairly non-trivial SDP relaxation of a *non-quadratic* formulation for bags of size 3. We analyze its rounding procedure using novel matrix decomposition techniques to obtain an algorithm which outputs an LTF satisfying at least $(1/12)$-fraction of the bags of size $\leq 3$. We also apply our techniques to bags of size $q \geq 4$ to provide a $\Omega(1/q)$-approximation guarantee for a weaker notion of satisfiability. We include comparative experiments on simulated data demonstrating the applicability of our algorithmic techniques.

From the complexity side we provide a hardness reduction to produce instances with bags of any constant size $q$. Our reduction proves the NP-hardness of satisfying more than $(1/q) + o(1)$ fraction of a satisfiable collection of such bags using as hypothesis any function of constantly many LTFs, showing thereby that the problem is harder to approximate as the bag size $q$ increases. Using a strengthened analysis, for $q = 2$ we obtain a $(4/9) + o(1)$ hardness factor for this problem, improving upon the $(1/2) + o(1)$ factor shown by [37].

## 1 Introduction

Our work studies the computational learnability of *linear threshold functions* (LTFs) in the *learning from label proportions* (LLP) framework, which is a generalization of traditional supervised learning. In this, a *bag* $B$ is a set of some (say $q$) feature vectors $\{\mathbf{x}_1, \ldots, \mathbf{x}_q\}$ with a corresponding $\{0, 1\}$-label proportion $\sigma_B \in [0, 1]$ implying that exactly $q\sigma_B$ out of the $q$ feature-vectors have 1 as their true label. Given a collection (or distribution) of $(B, \sigma_B)$ consistent with an unknown classifier, in LLP the goal is to fit a feature-vector level classifier hypothesis that matches the bag label proportions as closely as possible. One way to formalize this is by defining that a hypothesis classifier *satisfies* a bag $(B, \sigma_B)$ iff its predicted label proportion equals $\sigma_B$, with the goal being to maximize the number of bags satisfied by the hypothesis. This notion of satisfiability boils down to supervised learning when all bags are of size 1, and is a reasonable measure of classifier performance for small bags.

An LTF over $d$-dimensional feature-vectors $\mathbf{x}$ is given by $\mathsf{pos}(g(\mathbf{x}))$ for some linear function $g(x_1, \ldots, x_d) = \sum_{i=1}^{d} c_i x_i + c_{d+1}$, where $\mathsf{pos}(z) := \mathbb{1}_{\{z>0\}}$. Recently, [37] studied the *proper* LLP learnability of LTFs i.e, given a collection of bags and their label proportions consistent with an

unknown LTF, compute an LTF satisfying the maximum number of bags. It is well known ([7]) that in supervised learning (all bags of size 1) LTFs are learnable by LTFs (i.e., all bags can be satisfied) using linear programming. This however does not work for bags sizes $> 1$, and neither are random LTFs guaranteed to satisfy any significant fraction of the bags. The work of [37] studied this problem when all bags are of size $\leq 2$, giving an algorithm that satisfies at least $(2/5)$-fraction of all the bags, and $(1/2)$-fraction if all bags are *non-monochromatic* i.e., $\sigma_B \notin \{0, 1\}$ for all bags $B$. From the hardness side [37] showed that even on satisfiable instances where all bags are non-monochromatic of size 2, it is NP-hard to find an LTF satisfying more that $(1/2) + o(1)$ fraction of them.

The main algorithmic technique of [37] is based on the observation that the label proportion of a bag $B = \{\mathbf{x}_1, \mathbf{x}_2\}$ determines the sign of $g(\mathbf{x}_1)g(\mathbf{x}_2)$ where pos($g$) is a satisfying LTF with non-zero margin [1] i.e., $g(\mathbf{x}_1), g(\mathbf{x}_2) \neq 0$. Thus, one can write a collection of quadratic constraints over the coefficients of $g$. The corresponding semi-definite programming (SDP) relaxation can then be rounded using random hyperplanes to obtain the desired LTF.

However, the above approach is not directly applicable even for bags $B = \{\mathbf{x}_1, \mathbf{x}_2, \mathbf{x}_3\}$ of size 3 since their label proportions no longer determine the products $g(\mathbf{x}_i)g(\mathbf{x}_j)$ $(1 \leq i \neq j \leq 3)$. Therefore, the following question remained: *is there an efficient algorithm which given a collection of $(B, \sigma_B)$ s.t. $|B| \leq 3$ consistent with some LTF, computes an LTF that satisfies at least $\Omega(1)$-fraction of the bags*.

Our work answers the above question in the affirmative, using a fairly non-trivial SDP relaxation and new techniques to analyze the rounding algorithm. In particular, we show that if allowed the presence of certain boolean variables the problem admits a non-quadratic formulation which nevertheless can be relaxed to an SDP. For further analysis we prove a novel characterization of the condition $\mathbf{A} \succeq \mathbf{B}$ for two symmetric positive semi-definite (psd) matrices $\mathbf{A}$ and $\mathbf{B}$ in terms of their decomposition. Our algorithm provides an LTF satisfying at least $(1/12)$-fraction of the bags of size $\leq 3$. For bags of sizes $\geq 4$, we adapt this approach to provide a $\Omega(1/q)$-approximation for a weaker notion of bag satisfiability which is the same as satisfiability for monochromatic bags, but only requires *splitting* the non-monochromatic bags.

We also show hardness reduction to this problem for bags of any constant size $q \geq 2$. Unlike the reduction of [37], ours produces a mixture of non-monochromatic and monochromatic bags, and for general bag sizes $q \in \mathbb{Z}^+$ it yields a $(1/q) + (1)$ hardness factor for any boolean function of constantly many LTFs as hypothesis, providing evidence that the problem becomes harder as the bag size $q$ increases. For the specific case of $q = 2$ we obtain a hardness factor of $(4/9) + o(1)$ improving on the $(1/2) + o(1)$ bound of [37].

An overview of our algorithms, hardness result and their analysis is provided later in this section.

## 1.1 Previous Related Work

The study of LLP is motivated by applications in which only the aggregated labels for sets (bags) of feature vectors are available due to privacy or legal [35, 40] constraints or inadequate or costly supervision [13, 11]. LLP has been applied to several weakly supervised tasks, for e.g. IVF prediction [23] and image classification [8, 30]. Notably, small bag sizes – studied in this work – arise in real-world scenarios, e.g. [30] consider bags of size 50, and bag sizes $10 \sim 20$ are relevant for IVF applications (see Sec 1.2 of [4]).

There have been several works works applying a variety of techniques e.g. MCMC, clustering, linear classifiers, variants of SVM ([12, 22, 29, 35, 41], others ([33, 32, 39, 38] provided guarantees under distributional assumptions, while recent works [26, 15, 27] have proposed deep neural net based methods. There methods typically attempt to fit an ML model to a collection of bags and their label proportions by minimizing some loss between the label-proportions and the average model predictions, summed over all the bags. However, while being practically applicable, they do not provide any non-trivial worst case performance guarantees, even for learning LTFs in the LLP setting.

In contrast to the above, the study of computational learning in the LLP framework has been – apart from the work of [37] – fairly sparse. The LLP framework (as an analogue of PAC learning) was first formalized in the work of [42]. They bounded the generalization error of a trained classifier when taking the (bag, label-proportion)-pairs as instances sampled iid from some distribution. Their loss

---

[1]It is easy to see that the non-zero margin property can be assumed for a finite set of linearly separable points (see Lemma 2.1 of [37])

function was different – a weaker notion than the strict bag satisfaction predicate that [37] and our work use.

As mentioned, LTFs [7] are well known to be properly learnable without any distributional assumptions. In the presence of adversarial label noise however the problem is NP-hard even to approximate [1, 5, 10] with the optimal $(1/2 + \varepsilon)$-factor hardness shown by [16, 20], and generalized by [6] to hold even for constant degree *polynomial thresholds* as hypotheses.

## 1.2 Problem Definition

For an integer $q$, an instance of LLP-LTF$[q]$ consists of $(\mathcal{X}, \mathcal{B} = \{B_\ell\}_{\ell=1}^m, \{\sigma_\ell\}_{\ell=1}^m)$ where $\mathcal{X} = \{\mathbf{x}_1, \dots, \mathbf{x}_n\} \subseteq \mathbb{R}^d$ is a set of feature-vectors, and $\mathcal{B} = \{B_1, \dots, B_m\} \subseteq 2^{\mathcal{X}}$ s.t. $|B_j| \leq q$, is a collection of bags each of size at most $q$. For each bag $B_\ell$ there is a number $\gamma_\ell$ which is the sum of the $\{0, 1\}$-labels of the vectors in the bag, satisfying $\gamma_\ell \in \{0, \dots, |B_\ell|\}$, with the label proportion given by $\sigma_\ell := \gamma_\ell/|B_\ell|$. When $\sigma_\ell \in \{0, 1\}$ then $B_\ell$ is said to be *monochromatic* i.e., bags which have same label (either $0$ or $1$) for all their feature-vectors. The remaining bags $B_\ell$ necessarily of size $> 1$ are called *non-monochromatic*.

A bag $B_\ell \in \mathcal{B}$ is *satisfied* by some $F : \mathcal{X} \to \{0, 1\}$ if $\left(\sum_{\mathbf{x} \in B_\ell} F(\mathbf{x})\right) = \gamma_\ell = \sigma_\ell |B_\ell|$. We say that a bag is *split* by $F$ if $\left(\sum_{\mathbf{x} \in B_\ell} F(\mathbf{x})\right) \in \{1, \dots, |B_\ell| - 1\}$, while it is *unsplit* by $F$ if the latter assigns the same label to all the vectors in the bag. We say that a bag $B_\ell$ is *weakly satisfied* by $F$ if (i) $B$ is monochromatic and is satisfied by $F$, or (ii) $B$ is non-monochromatic and is split by $F$. Note that weak satisfiability is implied by satisfiability.

An instance of LLP-LTF$[q]$ is said to be *satisfiable* if there exists an LTF that satisfies all the bags. It is said to be *weakly satisfiable* if the LTF weakly satisfies all the bags. The goal is to find an LTF that (weakly) satisfies the most bags.

**Choice of objective.** The satisfiability condition is a natural generalization of the "classification" objective in supervised learning in which a $\{0, 1\}$-labeled example is either classified correctly or incorrectly. For small-sized bags, it is also a reasonable approximation to objectives based on the deviation of $\left(\sum_{\mathbf{x} \in B_\ell} F(\mathbf{x})\right)$ from $\gamma_\ell$. More importantly, as we shall see later in this paper, the satisfiability objective allows for a compact and tractable SDP relaxation in which any feasible solution can be rounded to an LTF with (in expectation) a non-trivial approximation guarantee.

## 1.3 Our Results

Our algorithmic result for satisfiable LLP-LTF$[3]$ is as follows.

**Theorem 1.1.** *Let $\mathcal{I}$ be a satisfiable LLP-LTF$[3]$ instance with $m$ bags partitioned into $m_0$ monochromatic bags of size $\leq 2$, $m_1$ non-monochromatic bags of size $2$, $m_2$ monochromatic bags of bags of size $3$, and $m_3$ non-monochromatic bags of size $3$. Then, there is a randomized polynomial time algorithm which on input $\mathcal{I}$ produces and LTF that satisfies in expectation at least $\left((m_0/2 + m_2/4 + m_3/6)/2 + m_1/2\right)$ bags. In the worst case, (if $m = m_3$) the algorithm satisfies in expectation at least $(1/12)$-fraction of the bags.*

The following theorem states our hardness result for satisfiable LLP-LTF$[q]$ and the improved hardness for satisfiable LLP-LTF$[2]$.

**Theorem 1.2.** *For any $\ell \in \mathbb{Z}^+$ and constant $\zeta > 0$ it is NP-hard to find any boolean valued function $f$ of $\ell$ LTFs that satisfies more than $(1/q + \zeta)$-fraction of the bags of a satisfiable LLP-LTF$[q]$ instance. For $q = 2$ in particular, a strengthened result holds with a hardness factor of $(4/9 + \zeta)$.*

We also provide the following algorithm for weakly-satisfying bags of a weakly-satisfiable LLP-LTF$[q]$ instance for any $q \in \mathbb{Z}^+$.

**Theorem 1.3.** *Let $\mathcal{I}$ be a weakly-satisfiable LLP-LTF$[q]$ instance with $m$ bags. Then, there is a randomized polynomial time algorithm which on input $\mathcal{I}$ produces an LTF that weakly-satisfies in expectation at least $(c_0 m/q)$ bags for some absolute constant $c_0 > 0$.*

## 1.4 Overview of the Algorithm

First, observe that it is the non-monochromatic bags that make the LLP-LTF problem difficult, as one can simply use linear programming to find an LTF satisfying all the monochromatic bags. This LTF may however not satisfy even a single non-monochromatic bag.

Let us first see how the algorithm of [37] for satisfiable LLP-LTF[2] proceeds. Since we can always append a coordinate with 1 to all feature vectors, assume that the satisfying LTF is given by $\mathsf{pos}(\langle \mathbf{r}, \mathbf{x} \rangle)$ (where $\mathbf{r}$ is the normal vector of the separating hyperplane) with non-zero margin, the latter is possible by perturbing the LTF if necessary. For a bag $B = \{\mathbf{x}_1, \mathbf{x}_2\}$, $\langle \mathbf{r}, \mathbf{x}_1 \rangle \langle \mathbf{r}, \mathbf{x}_2 \rangle$ is either positive or negative depending on whether the bag is monochromatic or non-monochromatic. There is a straightforward relaxation of this quadratic program to an SDP - substitute $\mathbf{r}\mathbf{r}^\mathsf{T}$ with a symmetric psd matrix $\mathbf{R}$ and replace $\langle \mathbf{r}, \mathbf{x}_1 \rangle \langle \mathbf{r}, \mathbf{x}_2 \rangle$ by $\mathbf{x}_1^\mathsf{T}\mathbf{R}\mathbf{x}_2$. Solving this SDP and using the psd decomposition $\mathbf{R} = \mathbf{L}^\mathsf{T}\mathbf{L}$ one obtains the same sign pattern for $\langle \mathbf{L}\mathbf{x}_1, \mathbf{L}\mathbf{x}_2 \rangle$. Further, the non-zero margin property guarantees $\|\mathbf{L}\mathbf{x}\|_2^2 = \langle \mathbf{L}\mathbf{x}, \mathbf{L}\mathbf{x} \rangle = \mathbf{x}^\mathsf{T}\mathbf{R}\mathbf{x} > 0$ for all the feature vectors $\mathbf{x}$ of the instance. A standard hyperplane rounding of $\mathbf{L}\mathbf{x}$ and taking the best of the obtained LTF or its negation yields a random LTF that satisfies non-monochromatic bags with probability $1/2$ and the monochromatic ones with probability $1/4$.

Note that the above algorithm crucially hinges on the fact that the label proportion of the 2-sized bag determines the sign of $\langle \mathbf{r}, \mathbf{x}_1 \rangle \langle \mathbf{r}, \mathbf{x}_2 \rangle$. This clearly is no longer true for a non-monochromatic $B = \{\mathbf{x}_1, \mathbf{x}_2, \mathbf{x}_3\}$ of size 3, and therefore it doesn't seem possible to write an SDP relaxation with only terms of the form $\mathbf{x}_i^\mathsf{T}\mathbf{R}\mathbf{x}_j$ and solve for $\mathbf{R}$ as the relaxation of $\mathbf{r}\mathbf{r}^\mathsf{T}$. Nevertheless, we observe that at least one of the two products $\langle \mathbf{r}, \mathbf{x}_1 \rangle \langle \mathbf{r}, \mathbf{x}_j \rangle$ $(j = 2, 3)$ is negative. Let us define boolean variables $s_{\{i,j\}}$ to be indicator of the event that $\langle \mathbf{r}, \mathbf{x}_i \rangle \langle \mathbf{r}, \mathbf{x}_j \rangle < 0$. Then, we have the following valid inequalities:

$$s_{\{i,j\}} \left( \mathbf{x}_i^\mathsf{T}\mathbf{R}\mathbf{x}_j \right) \leq 0 \ \forall 1 \leq i < j \leq 3, \quad \text{and} \quad \sum_{j=2,3} s_{\{1,j\}} \geq 1.$$

Of course, such constraints do not yield an SDP or a convex program due to the presence of the unknown variables $s_{\{i,j\}}$ in products with $\mathbf{R}$.

The key step for obtaining an SDP is to relax $s_{\{i,j\}}\mathbf{R}$ to a symmetric psd matrix $\mathbf{R}^{\{i,j\}}$ with the constraint $\mathbf{R} \succeq \mathbf{R}^{\{i,j\}}$ which is valid since $s_{\{i,j\}} \in \{0, 1\}$. Now, the above two constraints can be rewritten as

$$\mathbf{x}_i^\mathsf{T}\mathbf{R}^{\{i,j\}}\mathbf{x}_j \leq 0 \ \forall 1 \leq i < j \leq 3, \quad \text{and} \quad \sum_{j=2,3} \mathbf{R}^{\{1,j\}} \succeq \mathbf{R}.$$

From the last constraint above, we have $\mathbf{x}_1^\mathsf{T}\mathbf{R}^{\{1,2\}}\mathbf{x}_1 + \mathbf{x}_1^\mathsf{T}\mathbf{R}^{\{1,3\}}\mathbf{x}_1 \geq \mathbf{x}_1^\mathsf{T}\mathbf{R}\mathbf{x}_1$, and assuming $\mathbf{x}_1^\mathsf{T}\mathbf{R}^{\{1,2\}}\mathbf{x}_1 \geq \mathbf{x}_1^\mathsf{T}\mathbf{R}^{\{1,3\}}\mathbf{x}_1$ WLOG we have

$$\mathbf{x}_1^\mathsf{T}\mathbf{R}^{\{1,2\}}\mathbf{x}_1 \geq \mathbf{x}_1^\mathsf{T}\mathbf{R}\mathbf{x}_1/2 \ (*) \quad \text{along with,} \quad \mathbf{x}_2^\mathsf{T}\mathbf{R}^{\{1,2\}}\mathbf{x}_1 \leq 0 \ (**).$$

The above suggests that the angle between $\mathbf{L}\mathbf{x}_1$ and $\mathbf{L}\mathbf{x}_2$ cannot be too small, where $\mathbf{R} = \mathbf{L}^\mathsf{T}\mathbf{L}$. Indeed, suppose for the moment that we could replace the LHS of the first inequality above with $\langle \mathbf{L}\mathbf{x}_1, \mathbf{z} \rangle$ and the LHS of the second inequality with $\langle \mathbf{L}\mathbf{x}_2, \mathbf{z} \rangle$ with the guarantee that $\|\mathbf{z}\|_2 \leq \|\mathbf{L}\mathbf{x}_1\|_2$. A simple calculation shows that the angle between $\mathbf{z}$ and $\mathbf{L}\mathbf{x}_1$ is at most $\pi/3$, while the angle between $\mathbf{z}$ and $\mathbf{L}\mathbf{x}_2$ is at least $\pi/2$, implying a lower bound of $\pi/6$ on the angle between $\mathbf{L}\mathbf{x}_1$ and $\mathbf{L}\mathbf{x}_2$. Thus, random hyperplane rounding will separate $\mathbf{L}\mathbf{x}_1$ and $\mathbf{L}\mathbf{x}_2$ with probability at least $1/6$, and the obtained LTF or its negation will satisfy the bag with probability at least $1/12$.

The only question that remains is whether such a $\mathbf{z}$ as assumed above exists. We answer this in the affirmative by proving (in Sec. 2.1) the following: given psd $\mathbf{A}$, $\exists \mathbf{L}$ s.t. $\mathbf{A} = \mathbf{L}^\mathsf{T}\mathbf{L}$, and for any psd $\mathbf{B}$ these two conditions are equivalent: (i) $\mathbf{A} \succeq \mathbf{B}$; and (ii) , $\exists \mathbf{C}$ s.t $\mathbf{B} = \mathbf{L}^\mathsf{T}\mathbf{C}$ and $\mathbf{A} \succeq \mathbf{C}^\mathsf{T}\mathbf{C}$. Moreover, $\mathbf{L}$ is efficiently obtained by the spectral decomposition of $\mathbf{A}$.

For our analysis, letting $\mathbf{A} = \mathbf{R}$ and $\mathbf{B} = \mathbf{R}^{\{1,2\}}$, we can take $\mathbf{z} = \mathbf{C}\mathbf{x}_1$, and the last implication of (ii) yields $\|\mathbf{L}\mathbf{x}_1\|_2 \geq \|\mathbf{z}\|_2$.

This decomposition characterization of $\mathbf{A} \succeq \mathbf{B}$ for psd $\mathbf{A}, \mathbf{B}$ seems novel to the best of the authors' knowledge, and may prove useful in other geometric and SDP rounding techniques. It is easy to see that (ii) $\Rightarrow$ (i). The proof of the other direction is based on a specific choice of $\mathbf{L}$ which yields the

decomposition of $\mathbf{B} = \mathbf{L}^\mathsf{T}\mathbf{C}$. To show $\mathbf{A} \succeq \mathbf{C}^\mathsf{T}\mathbf{C}$ we invoke a variant of *Schur complement* positive definiteness condition.

For monochromatic 3-sized bags we use a standard SDP relaxation and random hyperplane rounding analysis. The complete algorithm for LLP-LTF[3] and its analysis are provided in Sec. 3. We include in Sec. 5 an experimental validation of our algorithm for LLP-LTF[3] on simulated data, showing that our method outperforms random LTF classifier, especially in the small margin scenarios. In these scenarios, the LTF of our algorithm has high predictive accuracy on instance-level test data, demonstrating the practical applicability of our algorithmic methods.

### 1.4.1 LLP-LTF[q]

In Appendix G, we extend the above algorithm to weakly satisfy bags of a weakly satisfiable LLP-LTF[q] instance for $q \geq 4$. Such instances also admit an analogous analysis for non-monochromatic bags as above and we obtain $(*)$ and $(**)$ except with a factor of $1/(q-1)$ instead of $1/2$, yielding an $\Omega(1/q)$ probability for random hyperplane rounding splitting $q$-sized non-monochromatic bags. Our techniques are also applicable to the related *multiple instance learning* (MIL) [8] of LTFs and we include an explanation in Appendix L. Obtaining guarantees for satisfying non-monochromatic bags size of $q \geq 4$ seems to require qualitatively stronger geometric techniques and in Appendix H of the supplementary we describe the technical issues in more detail. We also provide in Appendix K similar (to the LLP-LTF[3] experiments) empirical evaluation of our weak-satisfaction algorithm for LLP-LTF[4]. Lastly, in Appendix M we discuss how previous works can be used to derive generalizations bounds for satisfying LLP-LTF[q] instances.

### 1.5 Overview of Hardness for LLP-LTF[q]

The hardness reduction uses the template of a *dictatorship test* (see Chap. 7 of [31], Sec. 2 of [18]) and combines it with a variant of the Label Cover problem [3, 21]. A dictatorship test over a domain $[M]$ produces an instance $\mathcal{I}$ of the target problem, in our case LLP-LTF[q], such that (i) (completeness) corresponding to each $i \in [M]$ there is an LTF satisfying all bags of $\mathcal{I}$, (ii) (soundness) an LTF that does not have any distinguished (relatively large) coefficients does not satisfy more than some $\gamma < 1$ fraction of the bags. The crux is to construct dictatorship tests with large completeness vs soundness gap i.e., small $\gamma$.

Fix any $r \in \{1, \ldots, q\}$ and consider the following distribution $D^r$ on bags of $q$ feature vectors $\mathbf{X}^{(1)}, \ldots, \mathbf{X}^{(q)} \in \mathbb{R}^M$, each bag with label proportion $r/q$. First, sample $\mathbf{Z} \in \mathbb{R}^{M \times q}$ so that each row of $\mathbf{Z}_i$ is sampled iid uniformly from the set of vectors in $\{0, 1, 2\}^q$ which have exactly one coordinate with 2, $(r-1)$ with 1 and rest 0. We derive the vectors $\mathbf{X}^{(1)}, \ldots, \mathbf{X}^{(q)}$ from $\mathbf{Z}$ as follows for each $j \in [q]$: if $Z_{ij}$ is 0 then set $X_i^{(j)} = 0$, if $Z_{ij}$ is 1 then set $X_i^{(j)} = \delta$. Independently for each $i$ where $Z_{ij} = 2$, set $X_i^{(j)} = \delta$ w.p. $(1 - \varepsilon)$, set $X_i^{(j)} = 1$ w.p. $\varepsilon/2$ and set $X_i^{(j)} = 2$ w.p. $\varepsilon/2$. Here $\delta$ is taken to be small depending on $M$ and $q$, while $\varepsilon$ is a small constant depending on $q$ but not on $[M]$.

Note that for any $i$, there exactly $r$ of the $q$ vectors $\mathbf{X}^{(1)}, \ldots, \mathbf{X}^{(q)}$ have non-zero entries in the $i$th coordinates. Thus, each coordinate yields an LTF $\mathsf{pos}(X_i)$ which satisfies all the bags. The dictatorship test and the completeness analysis are presented in Appendix D.

For the soundness analysis (Appendix F), consider any LTF given by $\mathsf{pos}(h(\mathbf{X}))$, such that it has no large coefficients. Observe that $\{h(\mathbf{X}^{(j)})\}_{j=1}^q$ are identically distributed but not necessarily independent, while conditioned on $\mathbf{Z}$ they are independent but not identical. Using a fairly involved analysis we show is that there is a fixed Gaussian distribution $N(\mu, \Sigma)$ (independent of the choice of $\mathbf{Z}$, r) such that with high probability over the choice of $\mathbf{Z}$ each of $\{h(\mathbf{X}^{(j)})\}_{j=1}^q$ are distributed close to $N(\mu, \sigma)$. In effect, this implies that the probability that the bag is satisfied is at most, $\Delta_{r,\alpha} + o(1)$, where $\Delta_{r,\alpha} := \binom{q}{r}\alpha^r(1 - \alpha)^{q-r}$, and $\alpha := \mathsf{E}[\mathsf{pos}(g)]$, $g \sim N(\mu, \sigma)$, whre $\mathsf{E}$ is the expectation operator.

The above invariance is obtained (in Appendix F.1) through the randomness induced by the *noise* coordinates in $\mathbf{X}^{(j)}$ for a given $j$ i.e, those $i$ for which $Z_{ij}$ is sampled to be 2, on which $X_i^{(j)}$ are independently sampled to be 1 or 2 w.p. $\varepsilon/2$ each. Due to their small magnitude the $\delta$-valued coordinates in $X_i^{(j)}$ can essentially be ignored. After estimating bounds on the conditional (on

**Z**) expectation and variance of $h(\mathbf{X}^{(j)})$ we apply the Berry-Esseen theorem to obtain the desired invariance.

In Appendix C.1 we use the trick of *folding* over a real subspace [25] to encode the Label Cover and combine the above dictatorship test only on the $[M]$ labels of the Label Cover vertices. This combination and the label decoding (in Appendix C.3) is along the lines as previous works e.g. by [25, 21]. In fact, we combine the Label Cover instance with $D^r$ on bags of size $q$ with label proportions $r/q$ for all $r \in \{1, \ldots, q\}$. We note that the noise coordinates are identically distributed in each $D^r$. Thus, we are able to use the same $\mu$ and $\Sigma$ for each $r$ to obtain the $\Delta_{r,\alpha} + o(1)$ bound for each $r$ with the same $\alpha$. If we weigh each of these distributions uniformly, using the easy derivation that $\sum_{r=1}^q \Delta_{r,\alpha} \leq 1$ for $\alpha \in [0,1]$, we obtain a $(1/q + o(1))$ factor hardness as shown in Sec. 4. For $q = 2$, we obtain in Appendix B a better $4/9 + o(1)$ factor using explicit calculations.

Like the reduction of [37], ours also works for functions of constantly many LTFs as hypotheses, requiring the application of the multi-dimensional version of Berry-Esseen theorem.

The approach of decoupling by conditioning on **Z** is similar in spirit to that followed by [37] though their reduction has boolean coordinates which does not readily admit generalizations to larger bag sizes $q$. The main contribution of our hardness result is the design and analysis a dictatorship test that works for all bag sizes $q$ yielding bag-distributions of specific label proportions $r/q$ ($r = 1, \ldots, q$) with random-threshold like soundness $\Delta_{r,\alpha} + o(1)$.

**Organization of the paper.** The next section provides some mathematical preliminaries and the proof of our novel characterization of $\mathbf{A} \succeq \mathbf{B}$ for psd matrices. The latter is used in the proof of Theorem 1.1 in Sec. 3 which provides and analyzes our algorithm $\mathcal{A}$ for LLP-LTF[3]. Sec. 5 presents an experimental evaluation of our algorithm on simulated data. In Sec. 4, Theorem 1.2 is derived from the statement of our hardness reduction whose proof is deferred to the Appendix C. The proof of Theorem 1.3 is also omitted and appears in Appendix G.

## 2 Preliminaries

We state a few well known facts about matrices.

The *pseudo-inverse* of a diagonal matrix $\mathbf{D} = \mathrm{Diag}(\sigma_1, \ldots, \sigma_r, 0 \ldots, 0)$ with top $r$ non-zero entries and the rest 0 is given by $\mathbf{D}^\dagger := \mathrm{Diag}(\sigma_1^{-1}, \ldots, \sigma_r^{-1}, 0 \ldots, 0)$. A symmetric matrix $\mathbf{A}$ has a decomposition $\mathbf{A} = \mathbf{U}\mathbf{D}\mathbf{U}^\mathsf{T}$ for some diagonal matrix $\mathbf{D}$ and orthonormal matrix $\mathbf{U}$ i.e., satisfying $\mathbf{U}\mathbf{U}^\mathsf{T} = \mathbf{U}^\mathsf{T}\mathbf{U} = \mathbf{I}$. The pseudo-inverse is $\mathbf{A}^\dagger = \mathbf{U}\mathbf{D}^\dagger\mathbf{U}^\mathsf{T}$.

**Definition 2.1** (see [28, 9]). *For a real symmetric $n \times n$ matrix $\mathbf{A}$, the following conditions are equivalent: (1) $\mathbf{A} \succeq \mathbf{0}$, i.e $\mathbf{A}$ is positive semi-definite (psd), (2) $\mathbf{U}\mathbf{A}\mathbf{U}^\mathsf{T} \succeq \mathbf{0}$ for all orthonormal matrices $\mathbf{U}$, (3) $\mathbf{x}^\mathsf{T}\mathbf{A}\mathbf{x} \geq 0$ for all $\mathbf{x} \in \mathbb{R}^n$, (4) $\mathbf{A} = \mathbf{U}\mathbf{D}\mathbf{U}^\mathsf{T}$ for some orthonormal $\mathbf{U}$ with $\mathbf{D}$ being a non-negative diagonal matrix(spectral decomposition), (5) all the principal minors of $\mathbf{A}$ have non-negative determinant.*

For any two matrices, the *Loewner order* is given by $\mathbf{A} \succeq \mathbf{B} \Leftrightarrow \mathbf{A} - \mathbf{B} \succeq \mathbf{0}$. The *square-root* of a non-negative diagonal matrix $\mathbf{D} = \mathrm{Diag}(\sigma_1, \ldots, \sigma_n)$ is $\mathbf{D}^{1/2} := \mathrm{Diag}(\sigma_1^{1/2}, \ldots, \sigma_n^{1/2})$. For a psd $\mathbf{A} = \mathbf{U}\mathbf{D}\mathbf{U}^\mathsf{T}$, the square root is $\mathbf{A}^{1/2} = \mathbf{U}\mathbf{D}^{1/2}\mathbf{U}^\mathsf{T}$. The following lemma, a variant of the the Schur-complement definiteness property, can be found on page 88 of [9], see also Thm. 4.3 of [17].

**Lemma 2.2.** *For any $n \times n$ matrices $\mathbf{A}, \mathbf{B}$ and $\mathbf{C}$ where $\mathbf{A}$ and $\mathbf{C}$ are symmetric, let $\mathbf{X} = \left(\begin{smallmatrix} \mathbf{A} & \mathbf{B} \\ \mathbf{B}^\mathsf{T} & \mathbf{C} \end{smallmatrix}\right)$. Then, $\mathbf{X} \succeq \mathbf{0} \quad \Rightarrow \quad \mathbf{A} - \mathbf{B}\mathbf{C}^\dagger\mathbf{B}^\mathsf{T} \succeq 0$.*

### 2.1 A characterization of $\mathbf{A} \succeq \mathbf{B}$ for psd matrices

We prove the following lemmas which are used in our algorithmic results.

**Lemma 2.3.** *Given a real symmetric psd matrix $\mathbf{A}$, $\exists \mathbf{L}$ s.t. $\mathbf{A} = \mathbf{L}^\mathsf{T}\mathbf{L}$ and the following are equivalent: (i) $\mathbf{A} \succeq \mathbf{B}$, and (ii) $\exists \mathbf{C}$ s.t. $\mathbf{B} = \mathbf{L}^\mathsf{T}\mathbf{C}$ and $\mathbf{A} \succeq \mathbf{C}^\mathsf{T}\mathbf{C}$, for any real symmetric psd matrix $\mathbf{B}$. Further, $\mathbf{L}$ can be efficiently obtained from the spectral decomposition of $\mathbf{A}$.*

*Proof.* It is easy to see that (ii) $\Rightarrow$ (i) as follows. Considering any vector $\mathbf{x}$ we have,

$$\|\mathbf{C}\mathbf{x}\|_2^2 = \mathbf{x}^\mathsf{T}\mathbf{C}^\mathsf{T}\mathbf{C}\mathbf{x} \leq \mathbf{x}^\mathsf{T}\mathbf{A}\mathbf{x} = \mathbf{x}^\mathsf{T}\mathbf{L}^\mathsf{T}\mathbf{L}\mathbf{x} = \|\mathbf{L}\mathbf{x}\|_2^2 \tag{1}$$

where we use $\mathbf{A} \succeq \mathbf{C}^\mathsf{T}\mathbf{C}$ and $\mathbf{A} = \mathbf{L}^\mathsf{T}\mathbf{L}$. Thus, using (1)

$$\mathbf{x}^\mathsf{T}\mathbf{B}\mathbf{x} = \mathbf{x}^\mathsf{T}\mathbf{L}^\mathsf{T}\mathbf{C}\mathbf{x} = \langle \mathbf{L}\mathbf{x}, \mathbf{C}\mathbf{x} \rangle \leq \|\mathbf{L}\mathbf{x}\|_2 \|\mathbf{C}\mathbf{x}\|_2 \leq \|\mathbf{L}\mathbf{x}\|_2^2 = \mathbf{x}^\mathsf{T}\mathbf{A}\mathbf{x}$$

Thus, (ii) $\Rightarrow$ (i). The reverse is proved in Lemma 2.4 along with the explicit formula for $\mathbf{L}$. $\qquad\square$

**Lemma 2.4.** *Let* $\mathbf{A}$ *and* $\mathbf{B}$ *be two real, symmetric, psd* $k \times k$ *matrices such that* $\mathbf{A} \succeq \mathbf{B}$ *(‡). Then, with the spectral decomposition* $\mathbf{A} = \mathbf{U}\mathbf{D}\mathbf{U}^\mathsf{T} = \mathbf{L}^\mathsf{T}\mathbf{L}$ *where* $\mathbf{U}$ *is orthonormal,* $\mathbf{D}$ *is non-negative diagonal and* $\mathbf{L} = \mathbf{D}^{1/2}\mathbf{U}^\mathsf{T}$, *there exists* $\mathbf{C}$ *such that (i)* $\mathbf{B} = \mathbf{L}^\mathsf{T}\mathbf{C}$, *and (ii)* $\mathbf{A} \succeq \mathbf{C}^\mathsf{T}\mathbf{C}$.

*Proof.* Let $\overline{\mathbf{C}} := \mathbf{U}^\mathsf{T}\mathbf{B}\mathbf{U}$ be symmetric psd (Defn. 2.1). Condition (‡) of the lemma implies,

$$\mathbf{D} - \overline{\mathbf{C}} = \mathbf{U}^\mathsf{T}\mathbf{A}\mathbf{U} - \mathbf{U}^\mathsf{T}\mathbf{B}\mathbf{U} \succeq \mathbf{0}. \tag{2}$$

Suppose that $\mathbf{D}$ has top $r$ diagonal elements positive and the rest zero. Then $\overline{\mathbf{C}}$ is zero outside of the top $r \times r$ submatrix. Otherwise, $\mathbf{D} - \overline{\mathbf{C}}$ will have nonzero entries $-\overline{C}_{ir'} = -\overline{C}_{r'i}$ in the $(i, r')$ and $(r', i)$ entries for some $r' > r$ and $i$. On the other hand, the diagonal entry at $(r', r')$ is $-\overline{C}_{r',r'} = 0$ since both $(\mathbf{D} - \overline{\mathbf{C}})$ and $\overline{\mathbf{C}}$ are psd and have non-negative diagonals, and thus the $2 \times 2$ principal minor of $\mathbf{D} - \overline{\mathbf{C}}$ given by the $i$th and $r'$th rows/columns has a negative determinant which contradicts Defn. 2.1.

Since $\overline{\mathbf{C}}$ is zero outside of the top $r \times r$ submatrix, letting $\mathbf{I}_r$ be diagonal matrix with ones in the top $k$ entries and zero otherwise we have,

$$\mathbf{U}^\mathsf{T}\mathbf{B}\mathbf{U} = \mathbf{I}_r\mathbf{U}^\mathsf{T}\mathbf{B}\mathbf{U} = \mathbf{D}^{1/2}\left(\mathbf{D}^{1/2}\right)^\dagger \overline{\mathbf{C}} \Rightarrow \mathbf{B} = \mathbf{U}\mathbf{D}^{1/2}\left(\mathbf{D}^{1/2}\right)^\dagger \overline{\mathbf{C}}\mathbf{U}^\mathsf{T} = \mathbf{L}^\mathsf{T}\left(\mathbf{D}^{1/2}\right)^\dagger \overline{\mathbf{C}}\mathbf{U}^\mathsf{T}$$

Letting $\mathbf{C} := \left(\mathbf{D}^{1/2}\right)^\dagger \overline{\mathbf{C}}\mathbf{U}^\mathsf{T}$ yields property (i) of the lemma. For the second property observe that,

$$\mathbf{C}^\mathsf{T}\mathbf{C} = \mathbf{U}\overline{\mathbf{C}}^\mathsf{T}\left(\mathbf{D}^{1/2}\right)^\dagger \left(\mathbf{D}^{1/2}\right)^\dagger \overline{\mathbf{C}}\mathbf{U}^\mathsf{T} = \mathbf{U}\overline{\mathbf{C}}\mathbf{D}^\dagger\overline{\mathbf{C}}\mathbf{U}^\mathsf{T}, \tag{3}$$

using which $\mathbf{A} \succeq \mathbf{C}^\mathsf{T}\mathbf{C} \Leftrightarrow \mathbf{U}^\mathsf{T}\mathbf{A}\mathbf{U} \succeq \mathbf{U}^\mathsf{T}\mathbf{C}^\mathsf{T}\mathbf{C}\mathbf{U} \Leftrightarrow \mathbf{D} \succeq \overline{\mathbf{C}}\mathbf{D}^\dagger\overline{\mathbf{C}} \Leftarrow \mathbf{X} \succeq \mathbf{0}$, where $\mathbf{X} = \left(\begin{smallmatrix} \mathbf{D} & \overline{\mathbf{C}} \\ \overline{\mathbf{C}}^\mathsf{T} & \mathbf{D} \end{smallmatrix}\right) = \left(\begin{smallmatrix} \mathbf{D} & \overline{\mathbf{C}} \\ \overline{\mathbf{C}} & \mathbf{D} \end{smallmatrix}\right)$, and the last implication follows from Lemma 2.2. It remains to show that $\mathbf{X} \succeq \mathbf{0}$. For this let $\mathbf{z} = (x_1, \ldots, x_k, y_1, \ldots, y_k)$, and $\mathbf{x} = (x_1, \ldots, x_k), \mathbf{y} = (y_1, \ldots, y_k)$. Then,

$$\mathbf{z}^\mathsf{T}\mathbf{X}\mathbf{z} = \mathbf{x}^\mathsf{T}\mathbf{D}\mathbf{x} + \mathbf{y}^\mathsf{T}\mathbf{D}\mathbf{y} + 2\mathbf{x}^\mathsf{T}\overline{\mathbf{C}}\mathbf{y} \tag{4}$$

Since $\overline{\mathbf{C}}$ is symmetric psd we can write it as $\mathbf{V}^\mathsf{T}\mathbf{V}$ so that

$$\mathbf{x}^\mathsf{T}\overline{\mathbf{C}}\mathbf{x} + \mathbf{y}^\mathsf{T}\overline{\mathbf{C}}\mathbf{y} + 2\mathbf{x}^\mathsf{T}\overline{\mathbf{C}}\mathbf{y} = \langle \mathbf{V}\mathbf{x}, \mathbf{V}\mathbf{x} \rangle + \langle \mathbf{V}\mathbf{y}, \mathbf{V}\mathbf{y} \rangle + 2\langle \mathbf{V}\mathbf{x}, \mathbf{V}\mathbf{y} \rangle = \|\mathbf{V}\mathbf{x} + \mathbf{V}\mathbf{y}\|_2^2 \geq 0 \tag{5}$$

Substituting $2\mathbf{x}^\mathsf{T}\overline{\mathbf{C}}\mathbf{y} \geq -\left(\mathbf{x}^\mathsf{T}\overline{\mathbf{C}}\mathbf{x} + \mathbf{y}^\mathsf{T}\overline{\mathbf{C}}\mathbf{y}\right)$ into the RHS of (4) we obtain,

$$\mathbf{z}^\mathsf{T}\mathbf{X}\mathbf{z} \geq \mathbf{x}^\mathsf{T}(\mathbf{D} - \overline{\mathbf{C}})\mathbf{x} + \mathbf{y}^\mathsf{T}(\mathbf{D} - \overline{\mathbf{C}})\mathbf{y} \geq 0 \tag{6}$$

by (2) which holds for any $\mathbf{z}$. Thus, $\mathbf{X}$ is psd which completes the proof. $\qquad\square$

# 3 Algorithm for LLP-LTF[3]

## 3.1 SDP Relaxation

We define two collections of constraints NOSPLIT and SPLIT for monochromatic and non-monochromatic bags of size 3 respectively in Fig. 1. For a satisfiable instance $\mathcal{I} = (\mathcal{X} = \{\mathbf{x}_1, \ldots, \mathbf{x}_n\} \subseteq \mathbb{R}^d, \mathcal{B} = \{B_\ell\}_{\ell=1}^m, \{\sigma_\ell\}_{\ell=1}^m)$ of LLP-LTF[3] let $\tilde{\mathbf{x}}_i \in \mathbb{R}^{d+1}$ be given by appending an extra 1-valued coordinate to $\mathbf{x}_i$ for $i \in [n]$. With this the corresponding SDP relaxation is given in Fig. 2, and it enforces NOSPLIT constraints for monochromatic bags of size 3 and those given by SPLIT for the non-monochromatic 3-sized bags. Constraints for margin and bags of size 2 are the same as in the algorithm of [37].

**Feasibility of** SDP-I. As discussed in Sec. 1.4, if $\mathsf{pos}(\langle \mathbf{r}, \tilde{\mathbf{x}} \rangle)$ is the satisfying LTF, then we can set $\mathbf{R} = \mathbf{r}\mathbf{r}^\mathsf{T}$ and $\mathbf{R}^{\{i,j\}} = \mathbf{R}$ if $\langle \mathbf{r}, \tilde{\mathbf{x}}_i \rangle\langle \mathbf{r}, \tilde{\mathbf{x}}_j \rangle < 0$ and $\mathbf{0}$ otherwise. The arguments for the margin and 2-sized bag constraints are same as those in Sec 2.1 of [37], and those for the 3-sized bag constraints are informally presented in Sec. 1.4. We defer the formal proof to Appendix A.

NoSplit$(\mathbf{u}_1, \mathbf{u}_2, \mathbf{u}_3, \mathbf{Q})$ :

$$\forall 1 \leq r < s \leq 3 : \mathbf{u}_r^\mathsf{T} \mathbf{Q} \mathbf{u}_s \ \geq \ 0 \quad (7)$$

Split$\left( \mathbf{u}_1, \mathbf{u}_2, \mathbf{u}_3, \mathbf{Q}, \mathbf{Q}^{\{1,2\}}, \mathbf{Q}^{\{2,3\}}, \mathbf{Q}^{\{1,3\}} \right)$ :

$$\forall 1 \leq r < s \leq 3 : \mathbf{u}_r^\mathsf{T} \mathbf{Q}^{\{r,s\}} \mathbf{u}_s \leq 0 \quad (8)$$

$$\forall 1 \leq r < s \leq 3 : \mathbf{Q} - \mathbf{Q}^{\{r,s\}} \succeq \mathbf{0} \quad (9)$$

$$\mathbf{Q}^{\{1,2\}} + \mathbf{Q}^{\{1,3\}} \succeq \mathbf{Q} \quad (10)$$

$$\mathbf{Q}^{\{1,2\}} + \mathbf{Q}^{\{2,3\}} \succeq \mathbf{Q} \quad (11)$$

$$\mathbf{Q}^{\{1,3\}} + \mathbf{Q}^{\{2,3\}} \succeq \mathbf{Q} \quad (12)$$

Figure 1: NoSplit and Split

Given $(\{\tilde{\mathbf{x}}_i\}_{i=1}^n, \{B_\ell\}_{\ell=1}^m, \{\sigma_\ell\}_{\ell=1}^m)$. Vars: real, symmetric psd $\mathbf{R}$, $\mathbf{R}^{\{i,j\}}$ $1 \leq i < j \leq n$, s.t.

$$\forall i \in [n] : \qquad \tilde{\mathbf{x}}_i^\mathsf{T} \mathbf{R} \tilde{\mathbf{x}}_i \ > \ 0 \quad (13)$$

$$\forall B_\ell = \{\mathbf{x}_i, \mathbf{x}_j\}, (i < j) :$$

$$\text{if } \sigma_\ell \in \{0,1\} : \qquad \tilde{\mathbf{x}}_i^\mathsf{T} \mathbf{R} \tilde{\mathbf{x}}_j \ \geq \ 0 \quad (14)$$

$$\text{if } \sigma_\ell \notin \{0,1\} : \qquad \tilde{\mathbf{x}}_i^\mathsf{T} \mathbf{R} \tilde{\mathbf{x}}_j \ \leq \ 0 \quad (15)$$

$$\forall B_\ell = \{\mathbf{x}_i, \mathbf{x}_j, \mathbf{x}_k\}, (i < j < k) :$$

$$\text{if } \sigma_\ell \in \{0,1\} : \text{NoSplit}(\tilde{\mathbf{x}}_i, \tilde{\mathbf{x}}_j, \tilde{\mathbf{x}}_k, \mathbf{R}) \quad (16)$$

$$\text{if } \sigma_\ell \notin \{0,1\} : \text{Split}(\tilde{\mathbf{x}}_i, \tilde{\mathbf{x}}_j, \tilde{\mathbf{x}}_k, \mathbf{R},$$

$$\mathbf{R}^{\{i,j\}}, \mathbf{R}^{\{j,k\}}, \mathbf{R}^{\{i,k\}}) \quad (17)$$

Figure 2: SDP-I

Algorithm $\mathcal{A}$. Input: satisfiable instance $\mathcal{I}$ of LLP-LTF[3].

1. For each $\mathbf{x} \in \mathbb{R}^d$ define $\tilde{\mathbf{x}} := (x_1, \ldots, x_d, 1) \in \mathbb{R}^{d+1}$.
2. Solve SDP-I (Fig. 2) for psd matrix $\mathbf{R} \in \mathbb{R}^{(d+1) \times (d+1)}$.
3. Let $\mathbf{R} = \mathbf{U} \mathbf{D} \mathbf{U}^\mathsf{T}$ be its spectral decomposition. Let $\mathbf{L} = \mathbf{D}^{1/2} \mathbf{U}^\mathsf{T}$ so that $\mathbf{R} = \mathbf{L}^\mathsf{T} \mathbf{L}$.
4. Sample $\mathbf{g}$ u.a.r from $N(0,1)^{d+1}$.
5. Define the linear form $h(\mathbf{x}) := \langle \mathbf{L} \tilde{\mathbf{x}}, \mathbf{g} \rangle$.
6. Let $h^* \in \{h, -h\}$ such that $\mathsf{pos}(h^*(.))$ satisfies more bags of $\mathcal{I}$. Output $\mathsf{pos}(h^*(.))$.

Figure 3: Algorithm $\mathcal{A}$ for LLP-LTF[3].

## 3.2 SDP Algorithm and analysis

Fig. 3 provides the algorithm $\mathcal{A}$ for the satisfiable LLP-LTF[3] instance $\mathcal{I}$. We have the following lemma for bags of size 3.

**Lemma 3.1.** *Consider the linear form $h$ obtained in Step 5 of $\mathcal{A}$ (Fig. 3). Then, the probability of a non-monochromatic 3-size bag being split by $\mathsf{pos}(h(.))$ is at least $1/6$, and that of a 3-sized monochromatic being unsplit by $\mathsf{pos}(h(.))$ is at least $1/4$.*

*Proof.* Let $B$ be a bag of size 3 and by relabeling WLOG we can assume that $B = \{\mathbf{x}_1, \mathbf{x}_2, \mathbf{x}_3\}$.

*Case: $B$ non-monochromatic.* Using (10) we have

$$\tilde{\mathbf{x}}_1^\mathsf{T} \left( \mathbf{R}^{\{1,2\}} + \mathbf{R}^{\{1,3\}} \right) \tilde{\mathbf{x}}_1 \geq \tilde{\mathbf{x}}_1^\mathsf{T} \mathbf{R} \tilde{\mathbf{x}}_1 = \|\mathbf{L} \tilde{\mathbf{x}}_1\|_2^2, \qquad (18)$$

where $\mathbf{L}$ is as defined in Step 3 of $\mathcal{A}$ (Fig. 3). By averaging and WLOG we can assume that $\tilde{\mathbf{x}}_1^\mathsf{T} \mathbf{R}^{\{1,2\}} \tilde{\mathbf{x}}_1 \geq \|\mathbf{L} \tilde{\mathbf{x}}_1\|_2^2 / 2$ and by applying Lemma 2.4 to the guarantee that $\mathbf{R} \succeq \mathbf{R}^{\{1,2\}}$ (from (9)) we obtain that there exists a matrix $\mathbf{C}$ s.t.,

$$\mathbf{R}^{\{1,2\}} = \mathbf{L}^\mathsf{T} \mathbf{C} \quad \Rightarrow \langle \mathbf{L} \tilde{\mathbf{x}}_1, \mathbf{C} \tilde{\mathbf{x}}_1 \rangle = \tilde{\mathbf{x}}_1^\mathsf{T} \mathbf{L}^\mathsf{T} \mathbf{C} \tilde{\mathbf{x}}_1 = \tilde{\mathbf{x}}_1^\mathsf{T} \mathbf{R}^{\{1,2\}} \tilde{\mathbf{x}}_1 \geq \|\mathbf{L} \tilde{\mathbf{x}}_1\|_2^2 / 2, \qquad (19)$$

and

$$\mathbf{R} \succeq \mathbf{C}^\mathsf{T} \mathbf{C} \quad \Rightarrow \quad \|\mathbf{C} \tilde{\mathbf{x}}_1\|_2^2 = \tilde{\mathbf{x}}_1^\mathsf{T} \mathbf{C}^\mathsf{T} \mathbf{C} \tilde{\mathbf{x}}_1 \leq \tilde{\mathbf{x}}_1^\mathsf{T} \mathbf{R} \tilde{\mathbf{x}}_1 = \|\mathbf{L} \tilde{\mathbf{x}}_1\|_2^2. \qquad (20)$$

Further, using (8)

$$\langle \mathbf{L} \tilde{\mathbf{x}}_2, \mathbf{C} \tilde{\mathbf{x}}_1 \rangle = \tilde{\mathbf{x}}_2^\mathsf{T} \mathbf{L}^\mathsf{T} \mathbf{C} \tilde{\mathbf{x}}_1 = \tilde{\mathbf{x}}_2^\mathsf{T} \mathbf{R}^{\{1,2\}} \tilde{\mathbf{x}}_1 = \tilde{\mathbf{x}}_1^\mathsf{T} \mathbf{R}^{\{1,2\}} \tilde{\mathbf{x}}_2 \leq 0. \qquad (21)$$

Eqn. (13) implies $\|\mathbf{L} \tilde{\mathbf{x}}_b\|_2 > 0$ $(b = 1, 2)$, and by (19) we also have $\|\mathbf{C} \tilde{\mathbf{x}}_1\|_2 > 0$. Define the unit vectors:

$$\mathbf{z}_0 := \mathbf{C} \tilde{\mathbf{x}}_1 / \|\mathbf{C} \tilde{\mathbf{x}}_1\|_2, \quad \mathbf{z}_1 := \mathbf{L} \tilde{\mathbf{x}}_1 / \|\mathbf{L} \tilde{\mathbf{x}}_1\|_2, \text{ and } \quad \mathbf{z}_2 := \mathbf{L} \tilde{\mathbf{x}}_2 / \|\mathbf{L} \tilde{\mathbf{x}}_2\|_2. \qquad (22)$$

From (19), (20) and (21) we obtain that $\langle \mathbf{z}_0, \mathbf{z}_1 \rangle \geq 1/2$, and $\langle \mathbf{z}_0, \mathbf{z}_2 \rangle \leq 0$. For $b = 0, 1$ we can write $\mathbf{z}_b = c_{b0}\mathbf{z}_0 + c_{b1}\mathbf{z}_b^\perp$ where $\|\mathbf{z}_b^\perp\|_2 = 1$ and $\mathbf{z}_b^\perp \perp \mathbf{z}_0$ so that $c_{b0}^2 + c_{b1}^2 = 1$. Note that $\langle \mathbf{z}_0, \mathbf{z}_1 \rangle \geq 1/2$ implies that $c_{10} \geq 1/2$ and therefore $|c_{11}| \leq \sqrt{3}/2$. Further, $\langle \mathbf{z}_0, \mathbf{z}_2 \rangle \leq 0$ implies that $c_{20} \leq 0$. Thus,

$$\langle \mathbf{z}_1, \mathbf{z}_2 \rangle \leq c_{10}c_{20} + |c_{11}||c_{21}| \leq -(1/2)|c_{20}| + \left( \sqrt{3}/2 \right) \cdot 1 \leq \sqrt{3}/2. \tag{23}$$

Thus, the angle between $\mathbf{L}\tilde{\mathbf{x}}_1$ and $\mathbf{L}\tilde{\mathbf{x}}_2$ is at least $\pi/6$. From standard facts on random hyperplane rounding (see Appendix A of [37]) it is easy to see that $\mathsf{pos}(h(\mathbf{x}_1)) \neq \mathsf{pos}(h(\mathbf{x}_2))$ with probability at least $(\pi/6)/\pi = 1/6$.

*Case: $B$ monochromatic.* In this case, (13), (7) guarantee that $\{\mathbf{L}\tilde{\mathbf{x}}_b \mid b = 1, 2, 3\}$ are non-zero vectors with pairwise non-negative inner products. It is a well known fact (see [19]) that such vectors can be rotated to be contained in a three-dimensional orthant (cone subtended by three coordinate rays). Thus, the probability that the bag is unsplit by $\mathsf{pos}(h(.))$ is at least the probability that the inner products of three orthonormal vectors with $\mathbf{g}$ (as chosen in Step 4 of $\mathcal{A}$) all have the same sign. Each of these three inner products is an independent standard Gaussian, so the latter probability is $1/4$. $\square$

Since our algorithm $\mathcal{A}$ when restricted bags of size $\leq 2$ is the same as that given by [37], we can reuse the following lemma which summarizes the analysis in Sec. 2 of [37].

**Lemma 3.2** (Sec. 2 of [37])**.** *Any monochromatic bag of size $\leq 2$ is unsplit by $\mathsf{pos}(h(.))$ with probability at least $1/2$. Any non-monochromatic bag $2$-sized bag is split by $\mathsf{pos}(h(.))$ with probability at least $1/2$. Further, $h(\mathbf{x}_i) \neq 0$ $(1 \leq i \leq n)$ w.p. 1.*

Assuming that $h$ does not vanish any $\mathbf{x}_i$ (which happens w.p. 1) we obtain the following properties. If a monochromatic bag is usplit by $\mathsf{pos}(h(.))$ then it is satisfied by exactly one of $\mathsf{pos}(h(.))$ and $\mathsf{pos}(-h(.))$. This also holds for any non-monochromatic bag of size 3 split by $\mathsf{pos}(h(.))$. On the other hand a non-monochromatic bag of size 2, if split by $\mathsf{pos}(h(.))$, is satisfied by both $\mathsf{pos}(h(.))$ and $\mathsf{pos}(-h(.))$. This, along with Step 6 of $\mathcal{A}$ completes the proof of Theorem 1.1.

An analysis of the time complexity of $\mathcal{A}$ (which is asymptotically dominated by the time taken to solve the SDP) is provided in Appendix I.

# 4 Hardness Result

The following theorem, whose proof is provided in Appendix C, states our detailed hardness result .

**Theorem 4.1.** *For positive integers constants $q > 1, \ell \geq 1$, and any constants $\zeta > 0$ and $\{p_r \geq 0\}_{r=1}^q$ s.t. $\sum_{r=1}^q p_r = 1$, given an instance $\mathcal{I}$ of $\mathsf{LLP\text{-}LTF}[q]$ with $p_r$ fraction of bags of size $q$ and label proportion $r/q$, for $r \in \{1, \ldots, q\}$, it is NP-hard to distinguish between the following cases:*

YES Case. *There is an LTF that satisfies all the bags of $\mathcal{I}$.*

NO Case. *Any $\{0, 1\}$-function $f$ of at most $\ell$ LTFs satisfies at most $\Delta_{q,p_1,\ldots,p_q} + \zeta$ fraction of the bags in $\mathcal{I}$ where $\Delta_{q,p_1,\ldots,p_q} := \max_{\alpha \in [0,1]} \left( \sum_{r=1} p_r \Delta_{q,r,\alpha} \right)$ and $\Delta_{q,r,\alpha} := \binom{q}{r}\alpha^r(1-\alpha)^{q-r}$.*

**Proofs of Theorem 1.2.** We apply Theorem 4.1 with $p_r = 1/q$ for $r \in [q]$. In the NO case, the total fraction of bags satisfied by $f$ is $\upsilon := \max_{\alpha \in [0,1]} \left( \frac{1}{q} \sum_{r=1}^q \Delta_{q,r,\alpha} \right) + \zeta$ for an arbitrarily small constant $\zeta > 0$. Observing that $\sum_{r=1}^q \Delta_{q,r,\alpha} \leq \sum_{r=0}^q \Delta_{q,r,\alpha} = (\alpha + (1-\alpha))^q = 1$, we obtain that $\upsilon \leq 1/q + \zeta$. This, along with the Yes case, proves Theorem 1.2 for $\mathsf{LLP\text{-}LTF}[q]$.

For the case of $q = 2$ we show (in Appendix B) that $\min_{p \in [0,1]} \max_{\alpha \in [0,1]} p\alpha^2 + 2(1-p)\alpha(1-\alpha) = 4/9$ to obtain a $4/9 + \zeta$ hardness factor.

# 5 Experimental Evaluation

We compare our algorithm ($\mathcal{A}$) to random LTF ($\mathcal{R}$) evaluated on 25 instances for each row of Table 1 giving the avg. % bags satisfied by each method, and the last two columns providing the accuracy on test dataset obtained by sampling a bag (same as the bag distribution) and sampling u.a.r. one of the three feature-vectors from the bag.

For each instance, $m$ bags (of 3 $d$-dim. vectors each) are sampled, where each is non-monochromatic w.p. $3/4$. The small and large margin cases are analogous to the correlated and uncorrelated cases in the experiments of [37], and we similarly follow a best-of 5-trials based rounding for $\mathcal{A}$ and best-of 5 u.a.r. LTFs or their complements for $\mathcal{R}$. We see that (i) $\mathcal{A}$ satisfies on avg. 80-97% of the bags in the small margin cases, vastly outperforming $\mathcal{R}$, the average feature-vector level test accuracy of the LTF produced by our algorithm is quite high: 96-98% for $d = 10$ and 85-90% for $d = 40$, while that of random LTF is rather low at around 50-55%. (ii) $\mathcal{A}$ also betters $\mathcal{R}$ in most of the large margin cases. Additional details are included in Appendix K which also provides similar experimental evaluation for weakly-satisfying LLP-LTF[4].

| $d$ | $m$ | $\mathcal{A}$ | $\mathcal{R}$ | $\mathcal{A}_{\text{test}}$ | $\mathcal{R}_{\text{test}}$ |
|---|---|---|---|---|---|
| | | (small margin) | | | |
| 10 | 50 | **93.0** | 6.3 | **96.2** | 51.4 |
| 10 | 100 | **97.1** | 4.4 | **98.6** | 53.2 |
| 40 | 50 | **81.9** | 4.8 | **85.4** | 51.8 |
| 40 | 100 | **83.0** | 4.4 | **90.1** | 52.0 |
| | | (large margin) | | | |
| 10 | 50 | **51.0** | 46.3 | **67.7** | 58.1 |
| 10 | 100 | **55.9** | 46.5 | **75.1** | 63.1 |
| 40 | 50 | **43.5** | 41.2 | 52.9 | **53.2** |
| 40 | 100 | 40.2 | **40.9** | **53.4** | 52.3 |

Table 1: Our alg. ($\mathcal{A}$) vs rand. LTF ($\mathcal{R}$). Bag size 3.

*Remark.* The SDP formulation in our experiments for 3-sized bags differs slightly from the one in Fig. 2 by using alternate valid constraints for non-monochromatic bags. In particular, instead of $\mathbf{x}_i^{\mathsf{T}}\mathbf{R}^{\{i,j\}}\mathbf{x}_j \leq 0$, $i \neq j \in \{1,2,3\}$ (as described in Sec. 1.4) we add $\mathbf{x}_i^{\mathsf{T}}\mathbf{R}^{\{i,j\}}\mathbf{x}_j + \mathbf{x}_i^{\mathsf{T}}\mathbf{R}^{\{i,k\}}\mathbf{x}_k < 0$ for each $\{i,j,k\} = \{1,2,3\}$. It is easy to see that the new inequalities imply that there is $i \in \{1,2,3\}$ such that for each $j \in \{1,2,3\} \setminus \{i\}$, $\mathbf{x}_i^{\mathsf{T}}\mathbf{R}^{\{i,j\}}\mathbf{x}_j < 0$. Using this condition the rest of the analysis can be done as before yielding the same approximation guarantee, while it provided better observed experimental performance. We defer a formal explanation to Appendix J.

## 6   Conclusions

Our work develops novel linear algebraic techniques to design and analyze a non-trivial SDP relaxation based $(1/12)$-approximation for satisfiable LLP-LTF[3], for which no previous algorithm (other than trivial or random LTF) was known. We also prove a $1/q + o(1)$ factor hardness for LLP-LTF[$q$] for all constant $q$, and a strengthened $4/9 + o(1)$ factor for $q = 2$, improving on the previous $1/2 + o(1)$ factor [37]. We extend our algorithm to bag sizes $q \geq 4$ for for weaker notion of bag-satisfiability, obtaining $\Omega(1/q)$-approximate algorithm.

Experiments on simulated data of 3-sized bags shows that our algorithm can provide substantially improved (over random LTFs) performance, both in terms of bag satisfiability as well as on feature-vector level test evaluation.

The main open question in this line of work is to develop algorithms for satisfiable LLP-LTF[$q$] for $q \geq 4$. Of course, learnability in the LLP setting can also be studied for other natural classifiers such as DNF formulas and decision trees.

Another interesting direction is to study variants of the bag satisfiabliltiy objective such as those which minimize the average deviation (according to some distance e.g. $\ell_1$ or $\ell_2^2$) between the given bag label proportions and those induced by the solution classifier.

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
