# Supplementary Appendix for:
# Algorithms and Hardness for Learning Linear Thresholds from Label Proportions

**Rishi Saket**
Google Research India
rishisaket@google.com

## A    Feasibility of SDP-I (Fig. 2)

A simple argument based on the satisfiability of the instance $\mathcal{I}$ (see Lemma 2.1 in [37]) along with the definition of $\tilde{\mathbf{x}}_i$ ($i \in [n]$) yields that there exists $\mathbf{r} \in \mathbb{R}^{(d+1)}$ s.t. $\langle \mathbf{r}, \tilde{\mathbf{x}}_i \rangle \neq 0$ (non-zero margin) for $1 \leq i \leq n$, and $\text{pos}\left(\langle \mathbf{r}, \tilde{\mathbf{x}}_i \rangle\right)$ is an LTF that satisfies all the bags of $\mathcal{I}$. We take $\mathbf{R} := \mathbf{r}(\mathbf{r})^\mathsf{T}$. Observing that $\langle \mathbf{r}, \tilde{\mathbf{x}}_i \rangle \langle \mathbf{r}, \tilde{\mathbf{x}}_j \rangle = \tilde{\mathbf{x}}_i^\mathsf{T} \mathbf{r}(\mathbf{r})^\mathsf{T} \tilde{\mathbf{x}}_j = \tilde{\mathbf{x}}_i^\mathsf{T} \mathbf{R} \tilde{\mathbf{x}}_j$ the non-zero margin property implies that (13) is satisfied. Further, we have the following.

(i) Let $B_\ell = \{\mathbf{x}_i, \mathbf{x}_j\}$ be a bag of size 2. If $B_\ell$ is monochromatic then the inner products $\langle \mathbf{r}, \tilde{\mathbf{x}}_i \rangle$ are of the same sign and otherwise are of different signs. Thus,

$$(\sigma_\ell \in \{0,1\}) \Rightarrow \langle \mathbf{r}, \tilde{\mathbf{x}}_i \rangle \langle \mathbf{r}, \tilde{\mathbf{x}}_j \rangle \geq 0, \quad \text{and} \quad (\sigma_\ell \notin \{0,1\}) \Rightarrow \langle \mathbf{r}, \tilde{\mathbf{x}}_i \rangle \langle \mathbf{r}^*, \tilde{\mathbf{x}}_j \rangle \leq 0. \tag{24}$$

From the above it follows that $\mathbf{R}$ satisfies (14) and (15).

(ii) Let $B_\ell = \{\mathbf{x}_i, \mathbf{x}_j, \mathbf{x}_k\}$ be a bag of size 3. If $B_\ell$ is monochromatic then all the pairwise products as above are non-negative i.e.,

$$\langle \mathbf{r}, \tilde{\mathbf{x}}_i \rangle \langle \mathbf{r}, \tilde{\mathbf{x}}_j \rangle \geq 0, \quad \langle \mathbf{r}, \tilde{\mathbf{x}}_j \rangle \langle \mathbf{r}, \tilde{\mathbf{x}}_k \rangle \geq 0 \text{ and,} \quad \langle \mathbf{r}, \tilde{\mathbf{x}}_k \rangle \langle \mathbf{r}, \tilde{\mathbf{x}}_i \rangle \geq 0, \tag{25}$$

implying that the collection of NOSPLIT constraints (16) given by (7) are satisfied by $\mathbf{R}$.
Lastly, suppose $B_\ell$ is non-monochromatic. Let

$$\mathbf{R}^{\{r,s\}} = \begin{cases} \mathbf{R} & \text{if } \langle \mathbf{r}, \tilde{\mathbf{x}}_r \rangle \langle \mathbf{r}, \tilde{\mathbf{x}}_s \rangle < 0 \\ \mathbf{0} & \text{otherwise} \end{cases} \quad \text{for } r, s \in [n], \ r < s. \tag{26}$$

The definition of $\mathbf{R}^{\{r,s\}}$ directly ensures that (8) and (9) are satisfied. Now, for every $t \in \{i,j,k\}$ there is a $t' \in \{i,j,k\}$ s.t. $\langle \mathbf{r}, \tilde{\mathbf{x}}_t \rangle \langle \mathbf{r}, \tilde{\mathbf{x}}_{t'} \rangle = \tilde{\mathbf{x}}_t^\mathsf{T} \mathbf{R} \tilde{\mathbf{x}}_{t'} < 0$. Letting $t''$ be the third index, this guarantees that $\mathbf{R}^{\{t,t'\}} + \mathbf{R}^{\{t,t''\}}$ is either $\mathbf{R}$ or $2\mathbf{R}$, and thus (10)–(12) are satisfied.

## B    Proof of Theorem 1.2 for $q = 2$

We wish to compute

$$\min_{p \in [0,1]} \max_{\alpha \in [0,1]} p\alpha^2 + 2(1-p)\alpha(1-\alpha) \tag{27}$$

Define:

$$\Gamma(p, \alpha) := p\alpha^2 + 2(1-p)\alpha(1-\alpha) \tag{28}$$

for $p, \alpha \in [0,1]$. It is easy to see that if we choose $p \geq 1/2$ then letting $\alpha = 1$ yields $\Gamma(p, \alpha) \geq 1/2$. Therefore, let us consider $0 \leq p \leq 1/2$. Differentiating w.r.t $\alpha$ we obtain,

$$\frac{\partial \Gamma(p,\alpha)}{\partial \alpha} = \alpha(6p - 4) + 2(1-p), \quad \text{and,} \quad \frac{\partial^2 \Gamma(p,\alpha)}{\partial^2 \alpha} = 6p - 4. \tag{29}$$

Thus, in the range $0 \leq p \leq 1/2$, $\frac{\partial^2 \Gamma(p,\alpha)}{\partial^2 \alpha} < 0$, therefore $\Gamma(p, \alpha)$ attains a maximum when $\frac{\partial \Gamma(p,\alpha)}{\partial \alpha} = 0$. Thus we set $\alpha(6p - 4) + 2(1 - p) = 0$, to obtain the root $\alpha^* = (1-p)/(2-3p)$. Substituting this value of $\alpha$ along with some calculations we obtain

$$\Gamma(p, \alpha^*) = \frac{(1-p)^2}{(2-3p)}.$$

Differentiating the above w.r.t $p$ we find that it is minimized at $p = p^* = 1/3$, at which $\alpha^* = 2/3$. Observing that $\Gamma(1/3, 2/3) = 4/9$ completes the analysis.

# C  Proof of Theorem 4.1

Our hardness result is via a reduction from the Smooth-Label-Cover problem defined below.

**Definition C.1.** *An instance of* Smooth-Label-Cover $\mathcal{L}(G(V,E), N, M, \{\pi^{e,v} \mid e \in E, v \in e\})$ *consists of a regular connected (undirected) graph $G(V,E)$ with vertex set $V$ and edge set $E$. Every edge $e = (v_1, v_2)$ is associated with projection functions $\{\pi^{e,v_i}\}_{i=1}^2$ where $\pi^{e,v_i} : [M] \to [N]$. A vertex labeling is a mapping defined on $L : V \to [M]$. A labeling $L$ satisfies edge $e = (v_1, v_2)$ if $\pi^{e,v_1}(L(v_1)) = \pi^{e,v_2}(L(v_2))$. The goal is to find a labeling which satisfies the maximum number of edges.*

The following theorem states the hardness of Smooth-Label-Cover and is proved in Appendix A of [21].

**Theorem C.2.** *There exists a constant $c_0 > 0$ such that for any constant integer parameters $Q, R \geq 1$, it is* NP-*hard to distinguish between the following two cases for a* Smooth Label Cover *instance* $\mathcal{L}(G(V,E), N, M, \{\pi^{e,v} \mid e \in E, v \in e\})$ *with $M = 7^{(Q+1)R}$ and $N = 2^R 7^{QR}$:*

- *(YES Case) There is a labeling that satisfies every edge.*
- *(NO Case) Every labeling satisfies less than a fraction $2^{-c_0 R}$ of the edges.*

*In addition, the instance $\mathcal{L}$ satisfies the following properties:*

- *(Smoothness) For any vertex $w \in V$, $\forall i, j \in [M]$, $i \neq j$, $\mathrm{Pr}_{e \sim w}[\pi^{e,w}(i) = \pi^{e,w}(j)] \leq 1/Q$, where the probability is over a randomly chosen edge incident on $w$.*
- *For any vertex $v$, edge $e$ incident on $v$, and any element $i \in [N]$, we have $|(\pi^{e,v})^{-1}(i)| \leq d := 4^R$; i.e., there are at most $d = 4^R$ elements in $[M]$ that are mapped to the same element in $[N]$.*
- *(Weak Expansion) For any $\delta > 0$, let $V' \subseteq V$ and $|V'| = \delta \cdot |V|$, then the number of edges among the vertices in $|V'|$ is at least $\delta^2|E|$.*

Theorem 4.1 directly follows from the following theorem.

**Theorem C.3.** *For positive integers constants $q > 1, \ell \geq 1$, and any constants $\zeta > 0$ and $\{p_r \geq 0\}_{r=1}^q$ s.t. $\sum_{r=1}^q p_r = 1$, there is a setting of $Q, R$ in Theorem C.2 such that there is a polynomial time reduction from the corresponding instance $\mathcal{L}$ of* Smooth-Label-Cover *to an instance $\mathfrak{I}$ of* LLP-LTF$[q]$ *with $p_r$ fraction of bags of size $q$ and label proportion $r/q$, for $r \in \{1, \ldots, q\}$ satisfying:*

- YES Case. *If $\mathcal{L}$ is a YES instance then there is an LTF that satisfies all the bags of $\mathfrak{I}$.*

- NO Case. *If $\mathcal{L}$ is a NO instance then for any $\{0, 1\}$-valued function $f$ of at most $\ell$ LTFs, satisfies at most $\Delta_{q,p_1,\ldots,p_q} + \zeta$ fraction of the bags in $\mathfrak{I}$ where*

$$\Delta_{q,p_1,\ldots,p_q} := \max_{\alpha \in [0,1]} \left( \sum_{r=1}^q p_r \Delta_{q,r,\alpha} \right) \quad \text{where} \quad \Delta_{q,r,\alpha} := \binom{q}{r} \alpha^r (1-\alpha)^{q-r}. \quad (30)$$

This section is devoted to the proof of Theorem C.3. Given $q, \ell$ we first set the parameters of the reduction as follows:

$$\varepsilon := \frac{\varepsilon_0}{10^{10q\ell}} \left( \frac{\zeta}{8q\ell} \right)^{32} \quad \tau := \varepsilon_0 \frac{\varepsilon^8}{2^8} \quad \delta := \varepsilon_0 \frac{\tau^4}{M^2} \quad Q := \frac{4}{\varepsilon^2} \left( \frac{16d^8\ell}{\tau^6} \right)^2 \quad (31)$$

where $\varepsilon_0 \in (0, 1)$ is a small enough absolute constant. Here $d := 4^R$ is the size of the pre-images of the projective constraints in Theorem C.2, and $R$ is the free parameter that will be chosen to be small enough. These parameter settings will be used till the end of Appendix F.

Let $\mathcal{L}$ be the Smooth-Label-Cover instance obtained from Theorem C.2 using the above parameter setting. Let us define the space of coordinates as $\mathbb{R}^{V_{\mathcal{L}} \times [M]}$, and for a vector $\mathbf{X}$ in this space let $\mathbf{X}_v$ be the $M$-dimensional restriction to the coordinates corresponding to $v$. First, we define in Fig. 4 an intermediate instance $\tilde{\mathfrak{I}}$ of LLP-LTF$[q]$ as a sampling procedure for bags of size $q$ and their label proportions. The probabilities $(p_1, \ldots, p_q)$ are only used to sample bags from different $D_X^r$.

1. Uniformly sample a vertex $v \in V_{\mathcal{L}}$.

2. Sample $r \in [q]$ with probability $p_r$.

3. With $\varepsilon, \delta$ as chosen in (31) sample a bag of $M$-dimensional vectors $\left(\mathbf{X}^{(1)}, \ldots, \mathbf{X}^{(q)}\right) \leftarrow D_X^r$ (Fig. 5).

4. Define $q$ vectors $\left\{\tilde{\mathbf{X}}^{(j)}\right\}_{j=1}^q$ each in $\mathbb{R}^{V_{\mathcal{L}} \times [M]}$ as:

$$\tilde{\mathbf{X}}_w^{(j)} = \begin{cases} \mathbf{X}^{(j)} & \text{if } w = v \\ \mathbf{0} & \text{otherwise,} \end{cases} \qquad \forall w \in V_{\mathcal{L}}, j \in [q]. \tag{32}$$

5. Output the bag $\tilde{B} = \left\{\tilde{\mathbf{X}}^{(j)}\right\}_{j=1}^q$ with label proportion $\sigma_B = r/q$

Figure 4: Instance $\tilde{\mathfrak{I}}$

## C.1 Folding and Final Instance $\mathfrak{I}$

For any edge $e = (u,v) \in E_{\mathcal{L}}$ and element $j \in [N]$, define the vector $\mathbf{h}^{(e,j)} \in \mathbb{R}^{V_{\mathcal{L}} \times [M]}$ as follows,

$$h_{w,i}^{(e,j)} = \begin{cases} 1 & \text{if } w = u \text{ and } i \in (\pi^{e,u})^{-1}(j) \\ -1 & \text{if } w = v \text{ and } i \in (\pi^{e,v})^{-1}(j) \\ 0 & \text{otherwise.} \end{cases}$$

Therefore, for any vector $\tilde{\mathbf{X}} \in \mathbb{R}^{V_{\mathcal{L}} \times [M]}$,

$$\forall e = (u,v) \in E, \ j \in [N], \quad \tilde{\mathbf{X}} \perp \mathbf{h}^{(e,j)} \Leftrightarrow \sum_{i \in (\pi^{e,u})^{-1}(j)} \tilde{X}_{u,i} = \sum_{i' \in (\pi^{e,v})^{-1}(j)} \tilde{X}_{v,i'} \tag{33}$$

Define two subspaces $H$ and $F$ of $\mathbb{R}^{V_{\mathcal{L}} \times [M]}$ as:

$$H := \text{span}(\mathbf{h}^{(e,j)} \mid e \in E, \ j \in [N]\}) \quad \text{and } F = H^{\perp} \tag{34}$$

i.e, $F$ is the orthogonal complement of $H$ in $\mathbb{R}^{V_{\mathcal{L}} \times [M]}$

The final instance $\mathfrak{I}$ is obtained by replacing each bag $\tilde{B} = \left\{\tilde{\mathbf{X}}^{(j)}\right\}_{j=1}^q$ with a bag $\overline{B} = \left\{\overline{\mathbf{X}}^{(j)}\right\}_{j=1}^q$, where $\overline{\mathbf{X}}^{(j)}$ is the projection of the vector $\tilde{\mathbf{X}}^{(j)}$ onto $F$ and represented using a orthonormal basis for $F$ ($j \in [q]$). Thus, the entire instance $\mathfrak{I}$ along with the expected LTF solutions reside in $F$.

## C.2 Proof of YES Case

If $\mathcal{L}$ is a YES instance then consider a satisfying labeling $L : V_{\mathcal{L}} \to [M]$, and define the vector $\tilde{\mathbf{r}} \in \mathbb{R}^{V_{\mathcal{L}} \times [M]}$ as $\tilde{r}_{v,L(v)} = 1$ for all $v \in V_{\mathcal{L}}$ and all the other coordinates are 0. It is easy to see the $\tilde{\mathbf{r}} \perp H$ and therefore $\tilde{\mathbf{r}} \in F$. Letting $\overline{\mathbf{r}}$ be $\tilde{\mathbf{r}}$ written in the orthonormal basis of $F$ and consider the LTF $f^*(\overline{\mathbf{X}}) = \text{pos}\left(\langle \overline{\mathbf{r}}, \overline{\mathbf{X}} \rangle\right)$.

Observe that since $\tilde{\mathbf{r}} \perp H$, for any bag $\tilde{B} = \left\{\tilde{\mathbf{X}}^{(j)}\right\}_{j=1}^q$ from $\tilde{\mathfrak{I}}$ and the corresponding bag $\overline{B} = \left\{\overline{\mathbf{X}}^{(j)}\right\}_{j=1}^q$ of $\mathfrak{I}$,

$$\langle \overline{\mathbf{r}}, \overline{\mathbf{X}}^{(j)} \rangle = \langle \tilde{\mathbf{r}}, \tilde{\mathbf{X}}^{(j)} \rangle \tag{35}$$

Let $v \in V_{\mathcal{L}}$ chosen in Step 1 of Fig. 4 when $\tilde{B}$ was sampled in $\tilde{\mathfrak{I}}$. From the construction of the vectors in $\tilde{B}$ and the fact that $\text{pos}\left(\langle \mathbf{e}_{L(v)}, . \rangle\right)$ for the coordinate vector $\mathbf{e}_{L(v)} \in R^M$ satisfies all the bags of $D_X^r$ ($r \in [q]$) (See Appendix D.1) implies that the LTF $f^*$ satisfies all the bags of $\mathfrak{I}$.

## C.3 Proof of NO Case.

Consider any boolean function $f$ of $\ell$ LTFs over $F$ given by $\mathsf{pos}(h_s(\cdot)$ where $h_s(\cdot) := \langle \overline{\mathbf{c}}_s, \cdot \rangle + c_{s_0}$ ($s \in [q]$). By folding, the coefficient vectors are written in a basis for $F$, and therefore we can rewrite the coefficient vectors as $\mathbf{c}_s$ in $\mathbb{R}^{V_{\mathcal{L}} \times [M]}$ satisfying (33) ($s \in [q]$). Thus, in $\mathbb{R}^{V_{\mathcal{L}} \times [M]}$ the LTFs are given by $h_s(\tilde{X}) := \langle \mathbf{c}_s, \tilde{X} \rangle + c_{s_0}$.

Let us further for each $v \in V_{\mathcal{L}}$ define $h_{s,v}(\mathbf{X}) := \langle \mathbf{c}_{s,v}, \mathbf{X} \rangle + c_0$ over $\mathbb{R}^M$, for $s \in [\ell]$.

Suppose that $f(\mathsf{pos}(h_1), \ldots, \mathsf{pos}(h_\ell))$ satisfies more than $\Delta_{q,p_1,\ldots,p_q} + \zeta$ fraction of all the bags of $\mathfrak{I}$. Since $\overline{\mathbf{c}}_s \in F$ (35) for $\overline{\mathbf{r}}, \tilde{\mathbf{r}}$ is also satisfied for $\overline{\mathbf{c}}_s, \mathbf{c}_s$, i.e., the relevant inner products of the coefficient vectors and the bag feature vectors are preserved. Therefore, $f(\mathsf{pos}(h_1), \ldots, \mathsf{pos}(h_\ell))$ over $\mathbb{R}^{V_{\mathcal{L}} \times [M]}$ satisfies $\Delta_{q,p_1,\ldots,p_q} + \zeta$ fraction of all the bags of $\tilde{\mathfrak{I}}$.

By averaging, for $\zeta/2$ fraction of the vertices $v$ (call them nice), $f(\mathsf{pos}(h_1), \ldots, \mathsf{pos}(h_\ell))$ satisfies at least $\Delta_{q,p_1,\ldots,p_q} + \zeta/2$ fraction of the bags $\tilde{B}$ sampled after choosing $v$ in Step 1 of Fig. 4. By the construction, this implies that for nice vertices $v$ $f(\mathsf{pos}(h_{1,v}), \ldots, \mathsf{pos}(h_{\ell,v}))$ satisfies $\Delta_{q,p_1,\ldots,p_q} + \zeta/2$ fraction of the bags sampled from $D_{q,p_1,\ldots,p_q}$ where the latter samples bags from $D_X^r$ with probability $p_r$, for $r \in [q]$. By our setting of the parameters, this is contradicts the condition of Lemma F.3 so that for each nice $v$ we can select one $s_v \in [\ell]$ s.t. $\mathbf{c}_{s_v,v}$ is not $\tau$-regular (see Defn. E.7). Thus, by averaging there is one $s^* \in [\ell]$ s.t. $s_v = s^*$ for at least $1/\ell$ fraction of the nice vertices. Call these vertices *good*, and they therefore constitute $\zeta/(2\ell)$ fraction of all the vertices. For convenience let us abuse notation to let $\mathbf{c}_v$ denote $\mathbf{c}_{s^*,v}$.

Using the above each good $v$ define the subsets

$$S_0(v) := \{i \in [M] \mid |c_{v,i}| > \tau \|c_v\|_2\}, \tag{36}$$
$$S_1(v) := \{i \in [M] \setminus S_0(v) \mid |c_{v,i}| > (\tau/(4d))\|c_v\|_2\}, \tag{37}$$
$$S_2(v) := [M] \setminus (S_0(v) \cup S_1(v)). \tag{38}$$

Note that $1 \leq |S_0(v)| \leq 1/\tau^2$ and $|S_1(v)| \leq 16d^2/\tau^2$. By the density property of Theorem C.2 we have that the good vertices induce at least $(\zeta^2/(4\ell^2)$ fraction of the edges $E_{\mathcal{L}}$. Out of these call those edges $e = (u,v)$ *good* if it satisfies:

$$|\pi^{e,u}(S_0(u) \cup S_1(u))| = |S_0(u) \cup S_1(u)| \quad \text{and,} \quad |\pi^{e,v}(S_0(v) \cup S_1(v))| = |S_0(v) \cup S_1(v)| \tag{39}$$

Using the smoothness property from Theorem C.2, our choice of $Q$ along with an union bound of separating any pair in $S_0(u) \cup S_1(u)$ and similarly for $S_0(v) \cup S_1(v)$ we obtain that the good vertices induce at least $(\zeta^2/(8\ell^2))$ fraction of the edges in $E_{\mathcal{L}}$ as good edges.

We now show that,

$$\pi^{e,u}(S_0(u)) \cap \pi^{e,v}(S_0(v)) \neq \emptyset \quad \text{for any good edge } e = (u,v). \tag{40}$$

To see this, assume for a contradiction that a good edge $e = (u,v)$ has $\pi^{e,u}(S_0(u)) \cap \pi^{e,v}(S_0(v)) = \emptyset$, and assume WLOG that $\|\mathbf{c}_v\|_2 \geq \|\mathbf{c}_u\|_2$. Choose $i^* \in S_0(v)$ and let $j^* = \pi^{e,v}(i^*)$. By (39) $|(S_0(v) \cup S_1(v)) \cap (\pi^{e,v})^{-1}(j^*)| = 1$. Thus,

$$\left| \sum_{i \in \cap (\pi^{e,v})^{-1}(j^*)} c_{v,i} \right| \geq |c_{v,i^*}| - d\left((\tau/(4d))\|c_v\|_2\right) \geq (3\tau/4)\|c_v\|_2 \tag{41}$$

On the other hand, since $j^* \notin \pi^{e,u}(S_0(u))$, and $|S_1(u) \cap (\pi^{e,u})^{-1}(j^*)| \leq 1$ (by (39))

$$\left| \sum_{i \in \cap (\pi^{e,u})^{-1}(j^*)} c_{u,i} \right| \leq d\left((\tau/(4d))\|c_u\|_2\right) \leq (\tau/4)\|c_u\|_2 \tag{42}$$

However, (33) implies that the LHS of (41) and (42) should be the same, which from the above is a contradiction since we assumed $\|\mathbf{c}_v\|_2 \geq \|\mathbf{c}_u\|_2$.

Finally, we construct a good labeling by sampling a label u.a.r. from $S_0(v)$ for each good vertex $v$. By the size bound on $S_0(v)$, and (40), this procedure satisfies each good edge with probability at least $\tau^4$, thereby satisfying in expectation at least $\tau^4(\zeta^2/(8\ell^2))$ fraction of all the edges, which is a contradiction by a small enough setting of $R$ and the NO case of Theorem C.2.

## D  Dictatorship Test and Analysis

Let $M$ and $\ell$ be positive integers, and let $\delta \in (0, 1)$ be a parameter possibly depending on $M, \ell$ and $q$. Further, let $\varepsilon > 0$ be a constant to be chosen later (independent of $M$).

Let $X_1, \ldots, X_M$ be $M$ coordinates over which the dictatorship test will be created. First we define a distribution $\mathsf{D}_X^r$ bags each of size $q$ and observed label proportion $r/q$ i.e., with exactly $r$ 1s, for some $r \in \{1, \ldots, q\}$. For convenience let us define functions $\chi_s : \{0, 1, 2\}^q \to \mathbb{Z}^{\geq 0}$ for $s = 1, 2$ where $\chi_s(\mathbf{u})$ counts the number of $s$-valued coordinates in $\mathbf{u}$, for $s = 1, 2$. Let $\mathcal{U}^r$ be the uniform distribution over the subset $\{\mathbf{u} \in \{0, 1, 2\}^q \mid \chi_2(\mathbf{u}) = 1, \chi_1(\mathbf{u}) = r - 1\}$ i.e., the subset of vectors in $\{0, 1, 2\}^q$ having exactly one 2 and $(r - 1)$ 1 values. With this we define in Fig. 5 the distribution $D_X^r$.

---

Output: $q$ vectors $\left(\mathbf{X}^{(1)}, \ldots, \mathbf{X}^{(q)}\right)$, where $\mathbf{X}^{(j)} = (X_1^{(j)}, \ldots, X_M^{(j)})$, for $j \in [q]$.

1. Construct a matrix $\mathbf{Z} \in \{0, 1, 2\}^{M \times q}$ by independently for each $i \in [M]$ sampling $\mathbf{Z}_i$ uniformly from $\mathcal{U}^r$.

2. Sample $\tilde{\mathbf{Z}} \in \{0, 1, 2\}^{M \times q}$ by independently sampling each entry to be 0 w.p. $(1 - \varepsilon)$, 1 w.p. $\varepsilon/2$ and 2 w.p. $\varepsilon/2$.

3. For each $i \in [M], j \in [q]$ define

$$X_i^{(j)} := \begin{cases} 0 & \text{if } Z_{ij} = 0 \\ \delta & \text{if } Z_{ij} = 1 \\ \delta & \text{if } Z_{ij} = 2 \text{ and } \tilde{Z}_{ij} = 0 \\ 1 & \text{if } Z_{ij} = 2 \text{ and } \tilde{Z}_{ij} = 1 \\ 2 & \text{if } Z_{ij} = 2 \text{ and } \tilde{Z}_{ij} = 2. \end{cases} \tag{43}$$

4. Output $\left(\mathbf{X}^{(1)}, \ldots, \mathbf{X}^{(q)}\right)$ as a bag of exactly $r$ 1s.

---

Figure 5: Distribution $D_X^r, r \in \{1, \ldots, q - 1\}$

### D.1  Completeness Analysis

For any $i \in [M]$ consider the LTF given by $\mathsf{pos}(X_i)$. Consider $D_X^r$ wherein each row of $\mathbf{Z}$ is sampled from $\mathcal{U}^r$, i.e. having exactly $r$ non-zero values. By construction the coordinates of nonzero values in $(X_i^{(1)}, \ldots, X_i^{(q)})$ and in $\mathbf{Z}_i$ are the same. Therefore, $\mathsf{pos}(X_i)$ will classify as 1 exactly $r$ of the vectors in any bag sampled from $D_X^r$, thus satisfying all the bags of $D_X^r$.

The soundness analysis is fairly lengthy and tedious and appears in Appendix F.

## E  Preliminaries for Dictatorship Test Soundness Analysis

The following theorems are used for the soundness analysis of the dictatorship test in Appendix D

**Theorem E.1** (Berry-Esseen Theorem, [31]). *Let $X_1, \ldots, X_n$ be independent random variables with $\mathsf{E}[X_i] = 0$ and $\mathrm{Var}[X_i] = \sigma_i^2$. Let $\sigma^2 = \sum_{i \in n} \sigma_i^2$. Then,*

$$\sup_{t \in \mathbb{R}} \left| \Pr_{X_1, \ldots, X_n} \left[ \sigma^{-1} \sum_{i \in [n]} X_i \leq t \right] - \Phi(t) \right| \leq c\gamma \tag{44}$$

*where $c$ is a universal constant, $\Phi$ is the CDF of the standard Gaussian $N(0, 1)$, and $\gamma := \sigma^{-1} \max_{i \in [n]} (\mathsf{E}\left[|X_i|^3\right] / \sigma_i^2)$.*

**Theorem E.2** (Multi-dimensional Berry-Esseen Theorem, [34]). *Let $\mathbf{X}_1, \ldots, \mathbf{X}_n$ be independent random vectors in $\mathbb{R}^d$ with $\mathsf{E}[\mathbf{X}_i] = 0$. Let $\mathbf{S} = \sum_{i=1}^n \mathbf{X}_i$ and $\mathbf{\Sigma} = \mathrm{Cov}[\mathbf{S}]$. Then for all convex sets $A \subseteq \mathbb{R}^d$*

$$|\mathsf{P}\left[\mathbf{S} \in A\right] - \Pr[\mathbf{Z} \in A]| \leq Cd^{1/4}\gamma \tag{45}$$

*where $C$ is a universal constant, $\mathbf{Z} \sim N(0, \mathbf{\Sigma})$ and $\gamma := \sum_{i=1}^n \mathsf{E}\left[\left\|\mathbf{\Sigma}^{-1/2}\mathbf{X}_i\right\|_2^3\right]$.*

**Theorem E.3** (Chernoff-Hoeffding). *Let $X_1, \ldots, X_n$ be independent random variables, s.t. $a_i \leq X_i \leq b_i$, $\Delta_i = b_i - a_i$ for $i = 1, \ldots, n$. Then, for any $t > 0$,*

$$\Pr\left[\left|\sum_{i=1}^n X_i - \sum_{i=1}^n \mathsf{E}[X_i]\right| > t\right] \leq 2 \cdot \exp\left(-\frac{2t^2}{\sum_{i=1}^n \Delta_i^2}\right).$$

**Chebyshev's Inequality.** For any random variable $X$ and $t > 0$,

$$\Pr\left[|X| > t\right] \leq \mathsf{E}[X^2]/t^2. \tag{46}$$

The total variation distance between two distributions $P, Q$ over $\mathbb{R}^d$ is $\mathrm{TV}(P, Q) := \sup_{A \subseteq \mathbb{R}^d} |P(A) - Q(A)|$.

**Theorem E.4** ([14]). *If $\boldsymbol{\Sigma}_1$ and $\boldsymbol{\Sigma}_2$ are positive-definite $d \times d$ matrices, and let $\lambda_1, \ldots, \lambda_d$ be the eigenvalues of $\boldsymbol{\Sigma}_1^{-1}\boldsymbol{\Sigma}_2 - \mathbf{I}_d$, then for any $\boldsymbol{\mu} \in \mathbb{R}^d$*

$$\mathrm{TV}\left(N\left(\boldsymbol{\mu}, \boldsymbol{\Sigma}_1\right), N\left(\boldsymbol{\mu}, \boldsymbol{\Sigma}_2\right)\right) \leq (3/2)\left(\sum_{i=1}^d \lambda_i^2\right)^{1/2}$$

The total variation between one-dimensional Gaussians is as follows.

**Theorem E.5** ([14]).

$$\mathrm{TV}\left(N(\mu_1, \sigma_1^2), N(\mu_2, \sigma_2^2)\right) \leq \frac{3|\sigma_1^2 - \sigma_2^2|}{\sigma_1^2} + \frac{|\mu_1 - \mu_2|}{2\sigma_1}.$$

The following Gaussian anti-concentration bound (which is obtained by simple integration) showss that for $g \sim N(0, \sigma^2)$

$$\Pr[g \in [t, t + \delta]] \leq \delta/(\sigma\sqrt{2\pi}) \tag{47}$$

for any $t \in \mathbb{R}$ and $\delta > 0$. We will also use the following simple lemma which appears as Lemma B.8 in [37].

**Lemma E.6.** *Any boolean valued function $f$ over boolean variables $y_1, \ldots, y_\ell$ can be written as*

$$f(y_1, \ldots, y_\ell) = \sum_{H \subseteq [\ell]} \left(a_H \prod_{s \in H} y_s\right), \tag{48}$$

*where each $|a_H| \leq 2^\ell$ for each $H \subseteq [\ell]$.*

**Definition E.7** (Regularity). *A vector $\mathbf{c} \in \mathbb{R}^n$ is $\nu$-regular if $|c_i| \leq \nu\|\mathbf{c}\|_2$ for all $i \in [n]$.*

## F Dictatorship Test Soundness Analysis

In this section we shall consider $\ell$ linear forms $h_1 \ldots h_\ell : \mathbb{R}^M \to \mathbb{R}$ given by $h_s(\mathbf{X}) := \langle \mathbf{c}_s, \mathbf{X}\rangle + c_0$ such that $\mathbf{c}_s$ are unit and $\tau$-regular (Defn. E.7) vectors, for each $s \in [\ell]$. The choice of $\tau$ and other parameters is as given in (31). The following is the key lemma for our analysis, the proof of which we defer to Appendix F.1.

**Lemma F.1.** *There is a vector $\boldsymbol{\mu} \in \mathbb{R}^\ell$ and a covariance matrix $\boldsymbol{\Sigma} \in \mathbb{R}^{\ell \times \ell}$ satisfying: for any $r \in \{1, \ldots, q-1\}$ and any $j \in [q]$ and any subset $H \subseteq [\ell]$, with probability at least $(1 - \nu_0)$ over the choice of $\mathbf{Z}$ in Step 1 of $D_X^r$ (Fig. 5):*

$$\left|\Pr_{\mathbf{X}^{(j)} \leftarrow D_X^r}\left[\bigwedge_{s \in H}\left(h_s\left(\mathbf{X}^{(j)}\right) > 0\right)\right] - \Pr_{(g_1, \ldots, g_\ell) \leftarrow N(\boldsymbol{\mu}, \boldsymbol{\Sigma})}\left[\bigwedge_{s \in H}(g_s > 0)\right]\right| \leq \eta_0 \tag{49}$$

*where $\nu_0, \eta_0 = O(\ell^2 \varepsilon^{1/4})$.*

Let $f_1, \ldots, f_q$ be some boolean functions over $\ell$ boolean variables. With the setup as in Lemma F.1, define for convenience,

$$\widehat{y}_{js} := \mathsf{pos}\left(h_s\left(\mathbf{X}^{(j)}\right)\right), \tag{50}$$

for all $s \in [\ell]$ and $j \in [q]$ where $(\mathbf{X}^{(j)})_{j=1}^q$ is sampled from $D_X^r$ which is either specified or clear from context. Also, let $y_s := \mathsf{pos}(g_s)$ where $(g_1, \ldots, g_\ell) \leftarrow N(\boldsymbol{\mu}, \boldsymbol{\Sigma})$.

**Lemma F.2.** *With the variables defined as above,*

$$\left| \mathsf{E}_{D_X^r} \left[ \prod_{j=1}^q f_j \left( \widehat{y}_{j1}, \ldots, \widehat{y}_{j\ell} \right) \right] - \prod_{j=1}^q \mathsf{E}_{N(\boldsymbol{\mu}, \boldsymbol{\Sigma})} \left[ f_j \left( y_1, \ldots, y_\ell \right) \right] \right| \leq q\nu_0 + 2^{2q\ell} \cdot \eta_0, \qquad (51)$$

*for each $r \in [q-1]$.*

*Proof.* From Lemma E.6 we can write the boolean functions as $f_j(\omega_1, \ldots, \omega_\ell) :=$ $\sum_{H \subseteq [\ell]} a_{j,H} \prod_{s \in H} \omega_s$, where each $a_H$ is of magnitude at most $2^\ell$. Using this one can expand the first term of the LHS of (51) as:

$$\mathsf{E} \left[ \prod_{j=1}^q f_j \left( \widehat{y}_{j1}, \ldots, \widehat{y}_{j\ell} \right) \right] = \sum_{(H_j)_{j=1}^q \in \left( 2^{[\ell]} \right)^q} \mathsf{E} \left[ \prod_{j=1}^q \left( a_{j,H_j} \prod_{s \in H_j} \widehat{y}_{js} \right) \right]. \qquad (52)$$

From Lemma F.1 for all except $q\nu_0$ fraction of the choices of $\mathbf{Z}$, (49) holds for all $j \in [q]$.

Let us fix one such $Z$ and observe that this renders $\{\widehat{X}^{(j)}\}_{j=1}^q$ independent vectors. Applying (49) to the expectation on RHS of (52) we obtain,

$$\mathsf{E} \left[ \prod_{j=1}^q \left( a_{j,H_j} \prod_{s \in H_j} \widehat{y}_{js} \right) \mid \mathbf{Z} \right] = \prod_{j=1}^q \left( a_{j,H_j} \mathsf{E} \left[ \prod_{s \in H_j} \widehat{y}_{js} \mid \mathbf{Z} \right] \right) \qquad (53)$$

and

$$\left| \prod_{j=1}^q \left( a_{j,H_j} \mathsf{E} \left[ \prod_{s \in H_j} \widehat{y}_{js} \mid \mathbf{Z} \right] \right) - \prod_{j=1}^q \left( a_{j,H_j} \mathsf{E} \left[ \prod_{s \in H_j} y_s \mid \mathbf{Z} \right] \right) \right| \leq \prod_{j=1}^q \left| a_{j,H_j} \right| \eta_0 \leq 2^{q\ell} \eta_0. \qquad (54)$$

Applying the above two bounds to all the $2^{q\ell}$ terms in (52) and using the fact that the product of the boolean functions is at most one to bound the loss due to the error in the choice of $\mathbf{Z}$ to $q\nu_0$ we obtain (51). $\qquad \square$

We finally have the following lemma bounding the probability of satisfying bags sampled from $D_X^r$, $r \in \{1, \ldots, q\}$.

**Lemma F.3.** *Let $f$ be some a boolean function over $\ell$ boolean variables, such that $\alpha :=$ $\mathsf{E}[f(y_1, \ldots, y_s)]$. Then, then letting $\widehat{\Delta}$ be the probability that $f$ applied to the LTFs $\mathsf{pos}(h_s)$ $(s \in [\ell])$ satisfies the bags sampled from $D_X^r$ $(r \in [q])$ we have,*

$$\widehat{\Delta} \leq \binom{q}{r} \alpha^r (1 - \alpha)^{q-r} + \zeta_1, \qquad (55)$$

*where $\zeta_1 = O\left( (q\nu_0 + 2^{2ql}\eta_0) \cdot 2^{2q} \right)$.*

*Proof.* By definition we have that $\widehat{\Delta}$ is the expected sum of all terms $\prod_{j=1}^q f_j \left( \widehat{y}_{j1}, \ldots, \widehat{y}_{j\ell} \right)$ where exactly $r$ out of $\ell$ $f_j$s are $f$ and the rest $(1 - f)$. Applying (51) of Lemma F.2, and using the fact that $\alpha \in [0, 1]$ we obtain that the expectation of each such term is $\alpha^r (1 - \alpha)^{q-r} + (q\nu_0 + 2^{2ql}\eta_0) \cdot 2^q$. Multiplying this by the number $\binom{q}{r} \leq 2^q$ of the possible terms yield the bound of the lemma. $\qquad \square$

### F.1 Proof of Lemma F.1

Let us fix any $j \in [q]$ and $r \in [q-1]$ along with the choices of $\delta$ and $\tau$ as in (31).

### F.1.1 Conditional expectation of a single $h_s$

Consider the steps of $D_X^r$ in Fig. 5. Let $J := \{i \in [M] \mid Z_{ij} = 2\}$, and let $J_\varepsilon := \{i \in J \mid \tilde{Z}_{ij} \neq 0\}$. Note that $J$ is completely determined by $j$th column of $\mathbf{Z}$ while $J_\varepsilon$ depends on that along with the $j$th column of $\tilde{Z}$. For a given choice of $\mathbf{Z}$ in Step 1 of $D_X^r$ (Fig. 5) define the following quantities.

$$E_s^{(J)} := \mathsf{E}_{D_X^r}\left[h_s(X^{(j)}) \mid J\right], \qquad V_s^{(J)} := \mathrm{Var}_{D_X^r}\left[h_s(X^{(j)}) \mid J\right] \tag{56}$$

Fixing $J$, and randomizing $J_\varepsilon$, each coordinate $i$ in $J$ has $X_i^{(j)}$ $\delta$, 1 or 2 w.p. $1 - \varepsilon$, $\varepsilon/2$ and $\varepsilon/2$ respectively. Further, the coordinates outside $J$ are $\delta$ with probability $(r-1)/(q-1)$ and otherwise 0. Therefore,

$$E_s^{(J)} = \sum_{i \in [M]} \left(\mathbb{1}_{\{i \in J\}} \cdot E_{s,i,j} + (1 - \mathbb{1}_{\{i \in J\}}) \cdot \tilde{E}_{s,i,j}\right) + c_{s,0} \tag{57}$$

where

$$E_{s,i,j} = c_{s,i}\left(3\varepsilon/2 + \delta(1 - \varepsilon)\right), \qquad \tilde{E}_{s,i,j} = c_{s,i}\left((r-1)/(q-1)\right)\delta. \tag{58}$$

Thus, we can bound $\mathsf{E}_s^{(J)}$ as:

$$E_s^{(J)} \in \left[\overline{E}_s^{(J)} - \delta\|c_s\|_1, \overline{E}_s^{(J)} + \delta\|c_s\|_1\right], \tag{59}$$

where

$$\overline{E}_s^{(J)} := (3\varepsilon/2)\sum_{j \in J} c_{s,i} + c_{s,0} \quad \text{and} \quad \mathsf{E}_J\left[\overline{E}_s^{(J)}\right] = (3\varepsilon r/(2q))\sum_{j \in [M]} c_{s,i} + c_{s,0} \tag{60}$$

since $\mathbb{1}_{i \in J}$ are iid with $\Pr[i \in J] = 1/q$. Further,

$$\mathrm{Var}_J\left[\overline{E}_s^{(J)}\right] = \mathrm{Var}_J\left[(3\varepsilon/2)\sum_{j \in J} c_{s,i}\right] = (3\varepsilon/2)^2\left((1/q) - (1/q)^2\right)\|\mathbf{c}_s\|_2^2 \tag{61}$$

where the last equality uses the fact that each $i \in J$ independently w.p. $(1/q)$.

### F.1.2 Conditional covariance of $h_s, h_t$

Consider $s, t \in [\ell]$ (not necessarily distinct), and let us define:

$$C_{s,t,i} = \mathrm{Cov}\left(c_{s,i}X_{i,j}, c_{t,i}X_{i,j} \mid i \in J\right) \quad \text{and,} \quad \tilde{C}_{s,t,i} = \mathrm{Cov}\left(c_{s,i}X_{i,j}, c_{t,i}X_{i,j} \mid i \notin J\right) \tag{62}$$

We can compute the above quantities as:

$$C_{s,t,i} = \mathsf{E}\left[c_{s,i}c_{t,i}X_{i,j}^2\right] - E_{s,i,j}E_{t,i,j} \tag{63}$$

and a few calculations along with our setting of $\delta$ yield:

$$\left|C_{s,t,i} - \left[(5\varepsilon)/2 - (9\varepsilon^2/4)\right]c_{s,i}c_{t,i}\right| \leq \sqrt{\delta}\,|c_{s,i}c_{t,i}| \tag{64}$$

Further, it is easy to see that

$$\tilde{C}_{s,t,i} \leq \delta^2\,|c_{s,i}c_{t,i}|. \tag{65}$$

Therefore,

$$C_{s,t}^{(J)} := \mathrm{Cov}\left(h_s(\mathbf{X}^{(j)}, h_t(\mathbf{X}^{(j)}) \mid J\right), \quad \overline{C}_{s,t}^{(J)} := \left[(5\varepsilon)/2 - (9\varepsilon^2/4)\right]\sum_{i \in [M]} \mathbb{1}_{\{i \in J\}}c_{s,i}c_{t,i} \tag{66}$$

satisfy,

$$\left|C_{s,t}^{(J)} - \overline{C}_{s,t}^{(J)}\right| \leq 2\sqrt{\delta}\|\mathbf{c}_s\|_2\|\mathbf{c}_t\|_2. \tag{67}$$

where we use Cauchy-Schwatrz to bound $\sum_i |c_{s,i}c_{t,i}|$ by $\|\mathbf{c}_s\|_2\|\mathbf{c}_t\|_2$. We will now prove a two sided bound on $\overline{C}_{s,t}^{(J)}$. For this, observe that each $i \in J$ w.p. $(1/q)$ which implies that,

$$\mathsf{E}_J\left[\overline{C}_{s,t}^{(J)}\right] = \left[(5\varepsilon)/2 - (9\varepsilon^2/4)\right](1/q)\sum_{i \in [M]} c_{s,i}c_{t,i} \tag{68}$$

and furthermore,

$$\sum_{i\in[M]} (c_{s,i}c_{t,i})^2 \leq \left(\max_{i\in[M]} |c_{s,i}|\right)\left(\max_{i\in[M]}|c_{s,i}|\right)\sum_{i\in[M]}|c_{s,i}c_{t,i}| \quad \leq \quad \tau\|\mathbf{c}_s\|_2\tau\|\mathbf{c}_t\|_2 \cdot \|\mathbf{c}_s\|_2\|\mathbf{c}_t\|_2$$
$$= \quad \tau^2\|\mathbf{c}_s\|_2^2\|\mathbf{c}_t\|_2^2 \qquad (69)$$

using the fact that $\mathbf{c}_s$ and $\mathbf{c}_t$ are $\tau$-regular. Applying the Chernoff-Hoeffding bound (Theorem E.3) we obtain that

$$\Pr_J\left[\left|\overline{C}_{s,t}^{(J)} - \mathsf{E}_J\left[\overline{C}_{s,t}^{(J)}\right]\right| > \tau^{1/2}\|\mathbf{c}_s\|_2\|\mathbf{c}_t\|_2\right] \leq \exp(-1/\tau). \qquad (70)$$

which using (67) and the fact that $\mathbf{c}_s, \mathbf{c}_t$ are unit vectors,

$$\Pr_J\left[\left|C_{s,t}^{(J)} - \mathsf{E}_J\left[\overline{C}_{s,t}^{(J)}\right]\right| \leq \tau^{1/2}\right] \geq 1 - \exp(-1/\tau). \qquad (71)$$

### F.1.3   Berry-Esseen for single $h_s$

We now show using the above bounds that there is a fixed Gaussian distribution which is close to the distribution of $h_s(\mathbf{X}^{(j)})$ conditioned on $J$, for nearly all choices of $J$.

First we set $t = s$ and use (67), (68) and (70) and the setting of $\delta$ to obtain that except with probability $\exp(-1/\tau)$ over the choice of $J$,

$$\left|C_{s,s}^{(J)} - \mathsf{E}_J\left[\overline{C}_{s,s}^{(J)}\right]\right| \leq 2\tau^{1/2}\|\mathbf{c}_s\|_2^2, \text{ and, } \quad \mathsf{E}_J\left[\overline{C}_{s,s}^{(J)}\right] = \left[(5\varepsilon)/2 - (9\varepsilon^2/4)\right](1/q)\|\mathbf{c}_s\|_2^2. \quad (72)$$

implying (from our setting of $\tau$)

$$\left|C_{s,s}^{(J)} - \mathsf{E}_J\left[\overline{C}_{s,s}^{(J)}\right]\right| \leq 2\tau^{1/4}\mathsf{E}_J\left[\overline{C}_{s,s}^{(J)}\right]. \qquad (73)$$

Along with the second equality in (72), (61) and (59) (with our setting of $\varepsilon$) implies that,

$$\mathrm{Var}_J\left[\overline{E}_s^{(J)}\right] \leq \varepsilon\mathsf{E}_J\left[\overline{C}_{s,s}^{(J)}\right] \qquad (74)$$

Thus, using Chebyshev's inequality we obtain that

$$\Pr\left[\left|\overline{E}_s^{(J)} - \mathsf{E}_J\left[\overline{E}_s^{(J)}\right]\right| > \varepsilon^{1/4}\sqrt{\mathsf{E}_J\left[\overline{C}_{s,s}^{(J)}\right]}\right]$$
$$\leq \quad \Pr\left[\left|\overline{E}_s^{(J)} - \mathsf{E}_J\left[\overline{E}_s^{(J)}\right]\right| > \left(1/\varepsilon^{1/4}\right)\sqrt{\mathrm{Var}_J\left[\overline{E}_s^{(J)}\right]}\right]$$
$$\leq \quad \varepsilon^{1/4}. \qquad (75)$$

Combining the above with (59) and using the setting of $\delta$ which ensures (using the second equality in (72)) that

$$\delta\|c_s\|_1 \leq \varepsilon^{1/4}\sqrt{\mathsf{E}_J\left[\overline{C}_{s,s}^{(J)}\right]}$$

we obtain that except with probability $\exp(-1/\tau) + \varepsilon^{1/4}$ over the choice of $J$ that (73) holds along with

$$\left|E_s^{(J)} - \mathsf{E}_J\left[\overline{E}_s^{(J)}\right]\right| \leq \varepsilon^{1/4}\sqrt{\mathsf{E}_J\left[\overline{C}_{s,s}^{(J)}\right]} \qquad (76)$$

Fix one such choice "good' choice of $J$ guaranteed from above, and construct the sum

$$h_s(\mathbf{X}^{(j)}) - \mathsf{E}\left[h_s(\mathbf{X}^{(j)}) \mid J\right] = \sum_{i\in[m]} \Gamma_i, \quad \text{where} \quad \Gamma_i = c_{s,i}X_{ij} - \mathsf{E}\left[c_{s,i}X_{ij} \mid J\right]. \qquad (77)$$

From the above proved lower bounds on the variance of $h_s(\mathbf{X}^{(j)})$ conditioned on $J$, the $\tau$-regularity of $\mathbf{c}_s$ and our setting of $\delta$ and $\tau$, one upper can using straightforward calculations upper bound

$$\max_i \frac{\mathsf{E}[|\Gamma_i|^3 \mid J]}{\left(\sum_{k\in[M]}\mathsf{E}[|\Gamma_k|^2 \mid J]\right)^{1/2}\mathsf{E}[|\Gamma_i|^2 \mid J]}$$

by $\sqrt{\tau}$. One can then apply the Berry-Esseen theorem to obtain that for any $t \in \mathbb{R}$,

$$\left| \Pr\left[ h_s(\mathbf{X}^{(j)}) > t \mid J \right] - \Pr\left[ \tilde{\mathcal{G}} > t \right] \right| \leq O(\sqrt{\tau}) \tag{78}$$

where $\tilde{\mathcal{G}} \sim N\left( E_s^{(J)}, C_{s,s}^{(J)} \right)$. Moreover, from Theorem E.5 we obtain that for any $t \in \mathbb{R}$,

$$\left| \Pr\left[ \tilde{\mathcal{G}} > t \right] - \Pr\left[ \mathcal{G} > t \right] \right| \leq 6\tau^{1/4} + \varepsilon^{1/4} = O(\varepsilon^{1/4}), \tag{79}$$

where $\mathcal{G} \sim N\left( \mathsf{E}_J\left[ \overline{E}_s^{(J)} \right], \mathsf{E}_J\left[ \overline{C}_{s,s}^{(J)} \right] \right)$. Combining the above two, we obtain that,

$$\left| \Pr\left[ h_s(\mathbf{X}^{(j)}) > t \mid J \right] - \Pr\left[ \mathcal{G} > t \right] \right| \leq O(\varepsilon^{1/4}) \tag{80}$$

### F.1.4 Applying Mutlivariate Berry-Esseen Theorem

For applying the multivariate Berry-Esseen theorem we consider $h_1, \ldots, h_\ell$ together and construct their noisy versions as follows. We fix a choice of $J$ which is "good" (as defined in the previous subsection) for all $s \in [\ell]$, and also satisfies the constraint inside the probability in (71) for all $s, t\ in[\ell]$. This is true for all except $O(\ell^2 \varepsilon^{1/4})$ fraction of the choices of $J$. Let $\{\zeta_{s,i} \sim N\left( 0, \frac{\varepsilon^2}{64}|c_{s,i}|^2 \right)\}$ be mean-zero independent Gaussians independent of each other and independent of all other variables. Define for all $s \in [\ell]$

$$\tilde{h}_s(\mathbf{X}^{(j)}) = \sum_{i \in [M]} \left( c_{s,i} X_i^{(j)} + \zeta_{s,i} \right) + c_{s,0} \tag{81}$$

and note that the $\mathsf{E}[h_s \mid J] = \mathsf{E}[\tilde{h}_s \mid J]$ for all $J$. Further, $h_s - \tilde{h}_s$ is a mean-zero Gaussian with variance $(\varepsilon^2/64)$ which is at most $O(\varepsilon)$ fraction of $\mathsf{E}_J\left[ \overline{C}_{s,s}^{(J)} \right]$ and of $\overline{C}_{s,s}^{(J)}$ by our choice of $J$. Therefore using Thm. E.5,

$$\left| \Pr[h_s > t \mid J] - \Pr[\tilde{h}_s > t \mid J] \right| \leq O(\sqrt{\varepsilon}) \tag{82}$$

for any $t \in \mathbb{R}$ and $s \in [\ell]$. Now, define the vectors $\mathbf{Y}_1, \ldots, \mathbf{Y}_M \in \mathbb{R}^\ell$ as:

$$\mathbf{Y}_{i,s} := \left( c_{s,i} X_i^{(j)} + \zeta_{s,i} \right) - \mathsf{E}[c_{s,i} X_i^{(j)} \mid J] \tag{83}$$

Let $\mathbf{\Sigma}$ be the covariance matrix of given by the covariances $C_{s,t}^J$, and let $\tilde{\mathbf{\Sigma}}$ be the covariance matrix (conditioned on $J$) of $\mathbf{Y} := \sum_{i \in [M]} \mathbf{Y}_i$. Since $\|\mathbf{c}_s\|_2 = 1$ for $s \in [\ell]$, we have that

$$\tilde{\mathbf{\Sigma}} = \mathbf{\Sigma} + \frac{\varepsilon^2}{64}\mathbf{I}. \tag{84}$$

Using the bounds proved in the previous two subsections, the $\tau$-regularity of $\mathbf{c}_s$ ($s \in [\ell]$) and the choice of $\tau$, the following can be shown (analogous to Appendix D.5 of [37]):

$$\sum_{i \in [M]} \mathsf{E}\left[ \|\mathbf{Y}_i\|_2^3 \right] \leq O(\ell\sqrt{\tau}) \tag{85}$$

Using the fact that the minimum eigenvalue of $\tilde{\mathbf{\Sigma}}$ is at least $\varepsilon^2/64$, the maximum eigenvalue of $\tilde{\mathbf{\Sigma}}^{-1/2}$ is at most $8/\varepsilon$. Thus,

$$\sum_{i=1}^{M} \|\mathsf{E}\left[ \tilde{\mathbf{\Sigma}}^{-1/2}\mathbf{Y}_i \|_2^3 \right] \leq O\left( \ell\sqrt{\tau}/\varepsilon^3 \right). \tag{86}$$

Thus, applying the multivariate Berry-Esseen theorem:

$$|\Pr[\mathbf{Y} \in A \mid J] - \Pr[\mathbf{\Upsilon} \in A]| \leq O\left( \ell\sqrt{\tau}/\varepsilon^3 \right) \tag{87}$$

where $A$ is any convex subset of $\mathbb{R}^\ell$ and $\mathbf{\Upsilon} \sim N\left( \mathbf{0}, \tilde{\mathbf{\Sigma}} \right)$.

Now consider the matrix $\mathbf{\Sigma}'$ given by the entries $\mathsf{E}_J\left[ \overline{C}_{s,t}^{(J)} \right]$, and define $\overline{\mathbf{\Sigma}} := \mathbf{\Sigma}' + (\varepsilon^2/64)\mathbf{I}$. Note that $\overline{C}_{s,t}^{(J)}$ is the covariance $C_{s,t}^{(J)}$ obtained when $\delta = 0$, thereofore $\mathbf{\Sigma}'$ is an expectation over

$$\text{SPLIT}_t\left((\mathbf{u}_r)_{r=1}^t, \mathbf{Q}, (\mathbf{Q}^{\{r,s\}})_{1\leq r<s\leq t}\right):$$

$$\forall 1\leq r<s\leq t: \mathbf{u}_r^\mathsf{T}\mathbf{Q}^{\{r,s\}}\mathbf{u}_s \leq 0 \quad (89)$$

$$\forall 1\leq r<s\leq t: \mathbf{Q}-\mathbf{Q}^{\{r,s\}}\succeq \mathbf{0} \quad (90)$$

$$\forall r\in[t]: \sum_{s\in[t]\setminus\{r\}}\mathbf{Q}^{\{r,s\}}\succeq \mathbf{Q} \quad (91)$$

Figure 6: $\text{SPLIT}_t$ for $t\geq 3$

For bags non-monochromatic bags $B_\ell$ s.t. $|B_\ell|=t\in[4,q]$, $B_\ell=\{\mathbf{x}_{i_r}\}_{r=1}^t$,

$$\text{SPLIT}_t\left((\tilde{\mathbf{x}}_{i_r})_{r=1}^t, \mathbf{R}, (\mathbf{R}^{\{i_r,i_s\}})_{1\leq r<s\leq t}\right) \quad (92)$$

Figure 7: Constraints for non-monochromatic bags of size $t\in[3,q]$ in SDP-I-$q$.

covariance matrices and is therefore also a covariance matrix. Further, by the choice of $J$ satisfying the probability condition in (71) for all $s,t$, we obtain that

$$\|\tilde{\boldsymbol{\Sigma}}-\overline{\boldsymbol{\Sigma}}\|_F = \|\boldsymbol{\Sigma}-\boldsymbol{\Sigma}'\|_F \leq \ell^2\tau^{1/2}.$$

Using the fact that minimum eigenvalue of $\overline{\boldsymbol{\Sigma}}$ is at least $\varepsilon^{64}$, we obtain that the squares of the eigenvalues of $\overline{\boldsymbol{\Sigma}}^{-1}\left(\overline{\boldsymbol{\Sigma}}-\tilde{\boldsymbol{\Sigma}}\right)$ is at most $O(\ell^2\tau^{1/2}/\varepsilon^2)$. Also, we can add in the expectations to obtain from (87) and Theorem E.4

$$\left|\Pr\left[(\tilde{h}_1,\ldots,\tilde{h}_\ell)\in A \mid J\right] - \Pr[\overline{\boldsymbol{\Upsilon}}\in A]\right| \leq O\left(\ell^2\sqrt{\tau}/\varepsilon^3\right) \quad (88)$$

where $\overline{\boldsymbol{\Upsilon}}\sim N(\tilde{\boldsymbol{\mu}},\overline{\boldsymbol{\Sigma}})$ where $\tilde{\mu}_s=\mathsf{E}[\tilde{h}_s \mid J]=\mathsf{E}[h_s \mid J]$ for $s\in[\ell]$. From the choice of $J$ and we know that

$$|\tilde{\mu}_s-\mu_s|\leq \varepsilon^{1/4}\sqrt{\overline{\Sigma}_{s,s}},$$

where $\boldsymbol{\mu}$ is a expectation vector given by $\mu_s=\mathsf{E}_J\left[\overline{E}_s^{(J)}\right]$. This combined with (82) allows us to take $\boldsymbol{\mu}$ and $\overline{\boldsymbol{\Sigma}}$ with an $\nu_0,\eta_0=O(\ell^2\varepsilon^{1/4})$ in the statement of Lemma F.1.

# G   Proof of Thm. 1.3 : Weakly Satisfying LLP-LTF[$q$]

Our goal is to show an $\Omega(1/q)$ approximation for weakly satisfying bags of a weakly satisfiable LLP-LTF[$q$] instance, for all $q\geq 4$ with the case of $q\leq 3$ handled by Theorem 1.1. So, for the rest of this section we shall consider $q\geq 4$.

It is sufficient to provide an $\Omega(1/q)$-approximation for only non-monochromatic bags – if the instance containts more than $(1/q)$-fraction monochromatic bags, standard linear programming can be used to satisfy all of them (since weak satisfiability is same as satisfiability for monochromatic bags), otherwise the $\Omega(1/q)$ approximation can be applied to the $\geq(1-1/q)$-fraction of weakly satisfiable non-nonchromatic bags, together yielding an overall $\Omega(1/q)$-approximation. The rest of this section provides the $\Omega(1/q)$ approximation for weakly satisfying non-monochromatic bags of a weakly satisfiable LLP-LTF[$q$] instance $\mathcal{I}$.

The SDP relaxation has all the variables of SDP-I (Fig. 2) as well as all of its constraints for non-zero margin (i.e., (13)) and those for non-monochromatic bags of size 2. For non-monochromatic bags of size $t\in[3,q]$, Fig. 7 provides the constraints using $\text{SPLIT}_t$ (given in Fig. 6) which is a direct generalization of the corresponding formulation for bag size 3 in Fig. 1. We refer to the resultant SDP relaxation as SDP-1-$q$ (Fig. 7).

**Feasibility of** SDP-I-$q$. The feasibility follows from generalizing the arguments in Appendix A. We state them here for completeness. As before if the LTF given by $\mathsf{pos}(\langle\mathbf{r},\tilde{\mathbf{x}}\rangle)$ weakly satisfies $\mathcal{I}$ then we set $\mathbf{R}:=\mathbf{r}\mathbf{r}^\mathsf{T}$. Further the matrices $\mathbf{R}^{\{r,s\}}$ are as defined in (26).

To begin with, the feasibility of the non-zero margin constraints and those for non-monochromatic bags of size 2 is shown in Appendix A, noting that weak satisfiability by $\mathsf{pos}(\langle\mathbf{r},\tilde{\mathbf{x}}\rangle)$ is sufficient for the arguments to go through.

Consider a non-monochromatic bag $B_\ell=\{\mathbf{x}_{i_1},\ldots,\mathbf{x}_{i_t}\}$ of size $t\in[3,q]$. The definition of $\mathbf{R}^{\{r,s\}}$ (according to (26)) directly ensures that (89) and (90) are satisfied. By weak satisfiability, for every $r\in[t]$ there is a $s\in[t]$ s.t. $\langle\mathbf{r},\tilde{\mathbf{x}}_r\rangle\langle\mathbf{r},\tilde{\mathbf{x}}_s\rangle=\tilde{\mathbf{x}}_r^\mathsf{T}\mathbf{R}\tilde{\mathbf{x}}_s\leq 0$. Thus the LHS of (91) is $t'\mathbf{R}=t'\mathbf{Q}$ for some $t'\in[1,t-1]$, implying that (91) is satisfied.

Algorithm $\mathcal{A}_q$. Input: weakly satisfiable instance $\mathcal{I}$ of LLP-LTF[$q$].

1. For each $\mathbf{x} \in \mathbb{R}^d$ define $\tilde{\mathbf{x}} := (x_1, \ldots, x_d, 1) \in \mathbb{R}^{d+1}$.

2. Solve SDP-I-$q$ (as described in Sec. G) for psd matrix $\mathbf{R} \in \mathbb{R}^{(d+1)\times(d+1)}$.

3. Let $\mathbf{R} = \mathbf{U}\mathbf{D}\mathbf{U}^\mathsf{T}$ be its spectral decomposition. Let $\mathbf{L} = \mathbf{D}^{1/2}\mathbf{U}^\mathsf{T}$ so that $\mathbf{R} = \mathbf{L}^\mathsf{T}\mathbf{L}$.

4. Sample $\mathbf{g}$ u.a.r from $N(0,1)^{d+1}$.

5. Output the linear form $h(\mathbf{x}) := \langle \mathbf{L}\tilde{\mathbf{x}}, \mathbf{g} \rangle$.

Figure 8: Algorithm $\mathcal{A}_q$ for weakly satisfiable LLP-LTF[$q$].

## G.1 SDP Algorithm and analysis

Fig. 8 provides the algorithm $\mathcal{A}_q$ for the weakly satisfiable LLP-LTF[$q$] instance $\mathcal{I}$. We first observe that $\mathcal{A}_q$ is the same algorithm (up to Step 5. and a specific choice of $\mathbf{L}$) as that of [37], and by their analysis the probability that a non-monochromatic bag of size 2 is split by $\mathsf{pos}(h(.))$ is at least $1/2$. This, along with the following lemma implies that $\mathcal{A}_q$ weakly satisfies in expectation at least $1/(\pi(q-1))$ fraction of the non-monochromatic bags of the weakly satisfiable instance $\mathcal{I}$ of LLP-LTF[$q$] for $q \geq 4$. Along with the arguments in the beginning of this section, this completes the proof of Thm. 1.3.

**Lemma G.1.** *Consider the linear form $h$ obtained in Step 5 of $\mathcal{A}_q$ (Fig. 8). Then, the probability of a non-monochromatic $t$-sized bag ($t \in [3, q]$) being split by $\mathsf{pos}(h(.))$ is at least $1/(\pi(t-1))$.*

*Proof.* Let $B$ be a non-monochromatic bag of size $t \in [4, q]$ and by relabeling WLOG we can assume that $B = \{\mathbf{x}_1, \ldots, \mathbf{x}_t\}$. Using (91) we have

$$\tilde{\mathbf{x}}_1^\mathsf{T} \left( \sum_{2 \leq r \leq t} \mathbf{R}^{\{1,r\}} \right) \tilde{\mathbf{x}}_1 \geq \tilde{\mathbf{x}}_1^\mathsf{T}\mathbf{R}\tilde{\mathbf{x}}_1 = \|\mathbf{L}\tilde{\mathbf{x}}_1\|_2^2, \tag{93}$$

where $\mathbf{L}$ is as defined in Step 3 of $\mathcal{A}_q$ (Fig. 8). By averaging and WLOG we can assume that $\tilde{\mathbf{x}}_1^\mathsf{T}\mathbf{R}^{\{1,2\}}\tilde{\mathbf{x}}_1 \geq \|\mathbf{L}\tilde{\mathbf{x}}_1\|_2^2/(t-1)$ and by applying Lemma 2.4 to the guarantee that $\mathbf{R} \succeq \mathbf{R}^{\{1,2\}}$ (from (90)) we obtain that there exists a matrix $\mathbf{C}$ s.t.,

$$\mathbf{R}^{\{1,2\}} = \mathbf{L}^\mathsf{T}\mathbf{C} \quad \Rightarrow \langle \mathbf{L}\tilde{\mathbf{x}}_1, \mathbf{C}\tilde{\mathbf{x}}_1 \rangle = \tilde{\mathbf{x}}_1^\mathsf{T}\mathbf{L}^\mathsf{T}\mathbf{C}\tilde{\mathbf{x}}_1 = \tilde{\mathbf{x}}_1^\mathsf{T}\mathbf{R}^{\{1,2\}}\tilde{\mathbf{x}}_1 \geq \|\mathbf{L}\tilde{\mathbf{x}}_1\|_2^2/(t-1), \tag{94}$$

and

$$\mathbf{R} \succeq \mathbf{C}^\mathsf{T}\mathbf{C} \quad \Rightarrow \quad \|\mathbf{C}\tilde{\mathbf{x}}_1\|_2^2 = \tilde{\mathbf{x}}_1^\mathsf{T}\mathbf{C}^\mathsf{T}\mathbf{C}\tilde{\mathbf{x}}_1 \leq \tilde{\mathbf{x}}_1^\mathsf{T}\mathbf{R}\tilde{\mathbf{x}}_1 = \|\mathbf{L}\tilde{\mathbf{x}}_1\|_2^2. \tag{95}$$

Further, using (89)

$$\langle \mathbf{L}\tilde{\mathbf{x}}_2, \mathbf{C}\tilde{\mathbf{x}}_1 \rangle = \tilde{\mathbf{x}}_2^\mathsf{T}\mathbf{L}^\mathsf{T}\mathbf{C}\tilde{\mathbf{x}}_1 = \tilde{\mathbf{x}}_2^\mathsf{T}\mathbf{R}^{\{1,2\}}\tilde{\mathbf{x}}_1 = \tilde{\mathbf{x}}_1^\mathsf{T}\mathbf{R}^{\{1,2\}}\tilde{\mathbf{x}}_2 \leq 0. \tag{96}$$

Eqn. (13) implies $\|\mathbf{L}\tilde{\mathbf{x}}_b\|_2 > 0$ ($b = 1, 2$), and by (94) we also have $\|\mathbf{C}\tilde{\mathbf{x}}_1\|_2 > 0$. Define the unit vectors:

$$\mathbf{z}_0 := \mathbf{C}\tilde{\mathbf{x}}_1/\|\mathbf{C}\tilde{\mathbf{x}}_1\|_2, \quad \mathbf{z}_1 := \mathbf{L}\tilde{\mathbf{x}}_1/\|\mathbf{L}\tilde{\mathbf{x}}_1\|_2, \text{ and } \quad \mathbf{z}_2 := \mathbf{L}\tilde{\mathbf{x}}_2/\|\mathbf{L}\tilde{\mathbf{x}}_2\|_2. \tag{97}$$

From (94), (95) and (96) we obtain that $\langle \mathbf{z}_0, \mathbf{z}_1 \rangle \geq 1/(t-1)$, and $\langle \mathbf{z}_0, \mathbf{z}_2 \rangle \leq 0$. For $b = 0, 1$ we can write $\mathbf{z}_b = c_{b0}\mathbf{z}_0 + c_{b1}\mathbf{z}_b^\perp$ where $\|\mathbf{z}_b^\perp\|_2 = 1$ and $\mathbf{z}_b^\perp \perp \mathbf{z}_0$ so that $c_{b0}^2 + c_{b1}^2 = 1$. Note that $\langle \mathbf{z}_0, \mathbf{z}_1 \rangle \geq 1/(t-1)$ implies that $c_{10} \geq 1/(t-1)$ and therefore $|c_{11}| \leq \sqrt{1 - (t-1)^{-2}}$. Further, $\langle \mathbf{z}_0, \mathbf{z}_2 \rangle \leq 0$ implies that $c_{20} \leq 0$. Thus,

$$\begin{aligned}
\langle \mathbf{z}_1, \mathbf{z}_2 \rangle \leq c_{10}c_{20} + |c_{11}||c_{21}| &\leq -(1/(t-1))|c_{20}| + \left( \sqrt{1 - (t-1)^{-2}} \right) \cdot 1 \\
&\leq \sqrt{1 - (t-1)^{-2}} = \cos\theta,
\end{aligned} \tag{98}$$

where $\theta = \sin^{-1}(1/(t-1))$. Thus, the angle between $\mathbf{L}\tilde{\mathbf{x}}_1$ and $\mathbf{L}\tilde{\mathbf{x}}_2$ is at least $\theta$. From standard facts on random hyperplane rounding (see Appendix A of [37]) and the fact that $\sin(\gamma) \leq \gamma$ for $\gamma \in [0, \pi/2]$ it can be seen that $\mathsf{pos}(h(\mathbf{x}_1)) \neq \mathsf{pos}(h(\mathbf{x}_2))$ with probability at least $\theta/\pi \geq (1/(t-1))/\pi = 1/(\pi(t-1))$. $\qquad\square$

**Remark G.2.** *The relaxation SDP-1-q can be strengthened by also adding constraints (generalizing* NoSplit *from Fig. 1) for monochromatic bags. One could attempt an analysis similar to the monochromatic case of Lemma 3.1. However, the property that $t$ vectors with pairwise non-negative inner products can be rotated into a $t$-dimensional orthant is true in general only for $t \leq 4$ and counterexamples exist for $t \geq 5$ (see [19]). Therefore, a different (more tedious) analysis would be required for monochromatic bags of size $\geq 5$. In any case, this does not yield significant asymptotic improvement in the approximation guarantee, and we handle the monochromatic bags separately as mentioned in the beginning of this section.*

## H   Technical challenges in satisfying bags of size $\geq 4$

The algorithm for LLP-LTF[3] presented in Sec. 3 finds a linear threshold $\mathsf{pos}(h(.))$ which splits a non-monochromatic bag of size 3 with probability $1/6$, and then uses the crucial fact that one of $\mathsf{pos}(h(.))$ or $\mathsf{pos}(-h(.))$ satisfies the bag. This, however is not true for non-monochromatic bags of size $\geq 4$. For e.g. consider a bag $B_\ell = \{\mathbf{x}_i\}_{i=1}^4$ of size 4 such whose label proportion $\sigma_\ell = 1/2$. Now, if we apply algorithm $\mathcal{A}_q$ ($q = 4$) on the instance, $\mathsf{pos}(h(.))$ may satisfy $(1/4) \sum_{i \in [4]} \mathsf{pos}(h(\mathbf{x}_i)) \in \{1/4, 3/4\}$ in which case neither $\mathsf{pos}(h(.))$ nor $\mathsf{pos}(-h(.))$ satisfies $B_\ell$.

One may attempt to strengthen the $\mathrm{SPLIT}_t$ constraints of Fig. 6 depending on label proportion of the bag. However, this does not seem to resolve the problem since $(1/4) \sum_{i \in [4]} \mathsf{pos}(h(\mathbf{x}_i))$ depends on the combined geometry of $\{\mathbf{L}\mathbf{x}_i\}_{i=1}^4$. In the above example, it is possible that $\{\mathbf{L}\mathbf{x}_i\}_{i=1}^4$ are arranged colinearly on a long line segment with $\mathbf{L}\mathbf{x}_1$ and $\mathbf{L}\mathbf{x}_4$ located at the two ends, while $\mathbf{L}\mathbf{x}_2$ and $\mathbf{L}\mathbf{x}_3$ are both very close to the mid-point. It is easy to check that in such a situation, in the event that $\mathsf{pos}(h(.))$ splits the bag, it will w.h.p. have $1/4$ or $3/4$ as the label proportion since $\mathbf{L}\mathbf{x}_2$ and $\mathbf{L}\mathbf{x}_3$ will most likely not be separated while $\mathbf{L}\mathbf{x}_1$ and $\mathbf{L}\mathbf{x}_4$ will most likely be separated.

The above suggests that stronger constraints on the geometry of $\{\mathbf{L}\mathbf{x}_i\}_{i=1}^4$ may need to be derived to ensure that the algorithm satisfies non-monochromatic bags of size $\geq 4$.

## I   Time complexity analysis

We analyze the running time of the algorithm $\mathcal{A}$ given in Fig. 3. The time complexity is dominated by that of solving the SDP given in Fig. 2. This contains the following $(d + 1) \times (d + 1)$ variable matrices which are constrained to be psd:

- The matrix $\mathbf{R}$.
- For each 3-sized non-monochromatic bag $\{\mathbf{x}_i, \mathbf{x}_j, \mathbf{x}_k\}$:
    - The matrices $\mathbf{R}^{\{r,s\}}$ for distinct $r, s \in \{i, j, k\}$.
    - The three matrices $\overline{\mathbf{R}}^{\{r,s\}} := \mathbf{R} - \mathbf{R}^{\{r,s\}}$ for distinct $r, s \in \{i, j, k\}$.
    - The three matrices $\widehat{\mathbf{R}}^{\{r,s\}} := \mathbf{R}^{\{p,r\}} + \mathbf{R}^{\{p,s\}} - \mathbf{R}$, for each $p \in \{i, j, k\}$ with $\{r, s\} = \{i, j, k\} \setminus \{p\}$.

Let the total number of bags be $m$. Construct a matrix $\mathbf{Z}$ which has the above 9 matrices as block diagonal for each non-monochromatic bag, and an additional block diagonal for $\mathbf{R}$. Constraining $\mathbf{Z} \succeq 0$ ensures that all the block-diagonals are psd. The dimension $N$ of $\mathbf{Z}$ is at most $(9m(d + 1) + 1) \times (9m(d + 1) + 1)$. There are $M = O(md^2)$ linear constraints to ensure the consistency $\overline{\mathbf{R}}^{\{r,s\}}$ and $\widehat{\mathbf{R}}^{\{r,s\}}$ defined above with the matrices $\mathbf{R}^{\{r,s\}}$ and $\mathbf{R}$, and to ensure the constraints of the SDP.

Thus, we have a canonical SDP with a $N \times N$ psd matrix and $M$ constraints linear in its entries.

We can apply the algorithm of [24] which runs in time $\tilde{O}\left(\sqrt{N}\left(MN^2 + M^\omega + N^\omega\right)\right)$ where $\tilde{O}$ ignores the logarithmic factors and $\omega$ is the matrix multiplication exponent. For us this gives an $\tilde{O}\left(m^{3.5}d^{4.5} + m^{\omega+0.5}d^{2\omega+0.5}\right)$-time algorithm.

A similar running time analysis also works for the algorithm (in Appendix G) for weakly satisfying LLP-LTF[q], with both $N, M$ being multiplied by an $O(q^2)$ factor.

| $d$ | $m$ | $\mathcal{A}$ | $\mathcal{R}$ | $\mathcal{A}_{\text{test}}$ | $\mathcal{R}_{\text{test}}$ | $frac\_PC$ |
|---|---|---|---|---|---|---|
| | | | (small margin) | | | |
| 10 | 50 | **93.0** ±8.0 | 6.3 ±4.2 | **96.2** ±4.2 | 51.4 ±6.1 | 1.00 ±0.001 |
| 10 | 100 | **97.1** ±3.8 | 4.4 ±2.0 | **98.6** ±1.6 | 53.2 ±6.0 | 1.00 ±0.000 |
| 40 | 50 | **81.9** ±9.9 | 4.8 ±2.6 | **85.4** ±6.1 | 51.8 ±6.2 | 0.93 ±0.027 |
| 40 | 100 | **83.0** ±17.9 | 4.4 ±2.2 | **90.1** ±8.6 | 52.0 ±4.8 | 0.97 ±0.005 |
| | | | (large margin) | | | |
| 10 | 50 | **51.0** ±7.1 | 46.3 ±5.6 | **67.7** ±10.4 | 58.1±11.0 | 0.21 ±0.046 |
| 10 | 100 | **55.9** ±8.4 | 46.5 ±8.9 | **75.1** ±7.2 | 63.1±11.7 | 0.34 ±0.064 |
| 40 | 50 | **43.5** ±4.9 | 41.2 ±4.9 | 52.9 ±5.6 | **53.2**±5.5 | 0.04 ±0.002 |
| 40 | 100 | 40.2 ±3.4 | **40.9** ±3.6 | **53.4** ±5.7 | 52.3±6.7 | 0.04 ±0.002 |

Table 2: Our alg. ($\mathcal{A}$) vs rand. LTF ($\mathcal{R}$). Bag size 3.

## J  Alternate SDP constraints for non-monochromatic $3$-sized bags for LLP-LTF[3]

In the alternate SDP (as mentioned in the Remark in Sec. 5), we modify the SPLIT constraints in Fig. 1 by replacing (8) with the following:

$$\forall p \in \{1,2,3\}, r < s \in \{1,2,3\} \setminus \{p\} : \mathbf{u}_p^\mathsf{T} \mathbf{Q}^{\{p,r\}} \mathbf{u}_r + \mathbf{u}_p^\mathsf{T} \mathbf{Q}^{\{p,s\}} \mathbf{u}_s \; < \; 0. \tag{99}$$

To verify the feasibility of these constraints, let $B_\ell$, $\mathbf{r}$. $\mathbf{R}$ and $\mathbf{R}^{\{r,s\}}$ be as defined in Appendix A. Observe that when $B_\ell$ is a non-monochromatic bag, for any choice of $t \in \{i,j,k\}$ with $\{t, t', t''\} = \{1,2,3\}$, at least one of $\{\langle \mathbf{r}, \tilde{\mathbf{x}}_t \rangle \langle \mathbf{r}, \tilde{\mathbf{x}}_{t'} \rangle, \langle \mathbf{r}, \tilde{\mathbf{x}}_t \rangle \langle \mathbf{r}, \tilde{\mathbf{x}}_{t''} \rangle\} = \{\tilde{\mathbf{x}}_t^\mathsf{T} \mathbf{R} \tilde{\mathbf{x}}_{t'}, \tilde{\mathbf{x}}_t^\mathsf{T} \mathbf{R} \tilde{\mathbf{x}}_{t'}\}$ is negative. This, along with the definition of $\mathbf{R}^{\{r,s\}}$ in (26) implies (99).

The above described SDP variant combined with the same algorithm $\mathcal{A}$ (Fig 3) yields the same approximation factor as in Thm. 1.1 via a slight modification of the analysis in Sec. 3.2. In particular, for the proof of Lemma 3.1 we observe (using simple case analysis) that (99) implies the existence of $i \in \{1,2,3\}$ such that $\tilde{\mathbf{x}}_i^\mathsf{T} \mathbf{R}^{\{i,j\}} \tilde{\mathbf{x}}_j < 0$ for both values of $j \in \{1,2,3\} \setminus \{i\}$. Relabeling WLOG $i$ to 1 allows us to use the same proof of Lemma 3.1 as before. The rest of the analysis, and the approximation guarantee remain the same.

## K  Experiments: Additional Details

**Satisfiable** LLP-LTF[3]. Table 2 has the values with stddev error bars of the experiments discussed in Sec. 5. We also include the average *fractional principal component* $frac\_PC$ values. Here $frac\_PC$ is the ratio of the the maximum eigenvalue to the sum of eigenvalues of the solution $\mathbf{R}$ of the SDP for $\mathcal{A}$. For the small margin cases, this value is close to 1 indicating that $\mathbf{R}$ is nearly a rank-1 matrix whose principal component provides an LTF with excellent predictive performance. This further suggests the usefulness of our algorithm in small margin scenarios, which in traditional learning tasks are generally considered more difficult. Note that since these aggregates are over 25 independently sampled instances for each $(d, m)$, the significant standard deviations are not unexpected.

### K.1  Weakly Satisfying LLP-LTF[4]

We also conduct experiments on satisfiable LLP-LTF[4] instances (Table 3). The small and large margin cases are analogous to the 3-sized bags case, and each bag is non-monochromatic with probability $7/8$. We use $\mathcal{A}_q$ for $q = 4$ with the SDP constraints for monochromatic bags added in (see Remark G.2). Here, the bag-level performance in columns $\mathcal{A}_4$ and $\mathcal{R}$ is in terms of the % of weakly satisfied bags. We see similar trends as in the previous experiments – $\mathcal{A}_4$ substantially outperforms $\mathcal{R}$ in the small margin scenarios both at the bag level as well as on the instance-level test evaluation. The matrix $\mathbf{R}$ is also close to being rank-1 in the small margin scenarios. The performance is similar in the large margin cases, though $\mathcal{A}_4$ betters $\mathcal{R}$ in most large margin settings, especially in the evaluation on the instance-level test data.

| $d$ | $m$ | $\mathcal{A}_4$ | $\mathcal{R}$ | $\mathcal{A}_{4,\text{test}}$ | $\mathcal{R}_{\text{test}}$ | $frac\_PC$ |
|---|---|---|---|---|---|---|
| | | | (small margin) | | | |
| 10 | 50 | **89.1** ±5.9 | 53.8±5.8 | **83.3** ±25.2 | 49.7±6.00 | 0.99±0.002 |
| 10 | 100 | **91.1** ±3.9 | 51.5±3.4 | **93.5** ±5.3 | 49.1 ±4.9 | 0.99±0.001 |
| 40 | 50 | **86.2** ±8.0 | 53.8±4.5 | **71.8** ±18.2 | 51.6±6.1 | 0.86±0.038 |
| 40 | 100 | **86.8** ±8.7 | 51.0±4.3 | **82.0** ±8.6 | 51.2±5.1 | 0.92±0.008 |
| | | | (large margin) | | | |
| 10 | 50 | **83.8** ±4.3 | 82.6 ±5.4 | **61.0** ±13.5 | 54.52±10.6 | 0.23±0.054 |
| 10 | 100 | 82.9 ±3.4 | **83.0** ±3.3 | **65.88** ±15.8 | 54.32±12.5 | 0.34±0.109 |
| 40 | 50 | **83.5** ±3.6 | 83.2 ±3.6 | **54.3** ±10.3 | 53.4±7.9 | 0.04±0.003 |
| 40 | 100 | **81.8**±3.2 | 81.6±3.2 | 50.9±5.5 | **51.8** ±6.6 | 0.04±0.003 |

Table 3: Our alg. ($\mathcal{A}_4$) vs rand. LTF ($\mathcal{R}$). Bag size 4.

We note that in the small margin scenarios the instance level test data, the LTF produced by our algorithm generally has high accuracy, even though our SDP only optimizes for weak satisfiability of bags. Since weak satisfiability of non-monochromatic bags is invariant under negation of the classifier, the algorithm may sometimes find a negated version of a good classifier. Due to this, the standard deviation on test data of our algorithm on the test data is noticeably higher, especially in the small margin case.

### K.2 Compute Resources and Code

All the experiments were conducted on a 16 core Intel Xeon CPU 2.20GHz machine running Linux. The experimental code is available at:

https://github.com/google-research/google-research/tree/master/
Algorithms_and_Hardness_for_Learning_Linear_Thresholds_from_Label_Proportions.

## L  Multiple Instance Learning (MIL) of LTFs

In the MIL problem [8, 2], instead of label proportions we are only given the OR of the boolean labels of the feature-vectors of a bag i.e., an $\{0, 1\}$-*indicator* of the presence of at least one true label in each bag. Here, the bag-level objective is to find an LTF that MIL-*satisfies* most bags, where a bag is MIL-satisfied by an LTF if the OR of the LTF-induced labels matches the bag indicator.

Given an MIL instance with bags of size at most $q$ which is consistent with an (unknown) LTF, we can transform it into an instance of weakly-satisfiable LLP-LTF$[q]$ and then use the algorithm provided in Thm. 1.3 to obtain an approximate LTF solution which (in expectation) MIL-satisfies $\Omega(1/q)$-fraction of the bags. We formally prove the following theorem.

**Theorem L.1.** *Given an MIL instance with bags of size at most $q$ whose indicators are consistent with an (unknown) LTF, there is polynomial time randomized algorithm to produce an LTF that MIL-satisfies $(\gamma_0/q)$-fraction of the bags in expectation, where $\gamma_0 > 0$ is an absolute constant.*

*Proof.* Let the MIL instance have the collection of bags (each of size at most $q$) $\mathcal{B} = \mathcal{B}_0 \cup \mathcal{B}_1$ where $\mathcal{B}_b$ are the set of bags with indicator $b$, for $b \in \{0, 1\}$. Let $F^*$ be the unknown LTF that this consistent with the indicators of the bags. We can assume $q \geq 2$, otherwise we only have bags of size 1 and using linear programming we can MIL-satisfy all bags. We have the following cases.

Case 1: $F^*$ evaluates to 1 on all feature vectors present in the bags $\mathcal{B}_1$. Since $F^*$ necessarily evaluates to 0 on all feature-vectors in $\mathcal{B}_0$, in this case we know the labels of all feature vectors of the MIL instance. Thus, using linear programming we can find an LTF $F$ whose evaluation on the feature-vectors of the instance is identical to that of $F^*$ and thus MIL-satisfies all the bags.

Case 2: Case 1 does not hold and $|\mathcal{B}_0| \geq (1/q)|\mathcal{B}|$. In this case we can simply find (using linear programming) find $F$ which evaluates to 0 (same as $F^*$) on all feature-vectors of bags in $\mathcal{B}_0$ and thus MIL-satisfy all bags in $\mathcal{B}_0$.

Case 3: Case 1 does not hold and $|\mathcal{B}_0| < (1/q)|\mathcal{B}|$. In this case, we have $|\mathcal{B}_1| > (1 - 1/q)|\mathcal{B}| > |\mathcal{B}|/2$ (since $q \geq 2$), and there exists $\mathbf{x}^*$ in some bag of $\mathcal{B}_1$ s.t. $F^*(\mathbf{x}^*) = 0$. For a feature-vector $\mathbf{x}$, define an instance $\mathcal{I}(\mathbf{x})$ of LLP-LTF$[q + 1]$ with only non-monochromatic bags given by $\{\tilde{B} \mid \tilde{B} = B \cup \{\mathbf{x}\}, B \in \mathcal{B}_1\}$. The definitions of $\mathcal{B}_1$ and $\mathbf{x}^*$ imply that $\mathcal{I}(\mathbf{x}^*)$ is weakly-satisfiable by the LTF $F^*$. Using this we have the following algorithmic steps:

1. Find an $\mathbf{x}_0$ from the bags of $\mathcal{B}_1$ such that the SDP for the collection of non-monochromatic bags in $\mathcal{I}(\mathbf{x}_0)$ given in Fig. 7 is feasible for bag size $q + 1$. By the above there is one such $\mathbf{x}_0$ which can be found in polynomial time.

2. Use Algorithm $\mathcal{A}_{q+1}$ from Fig. 8 to obtain an LTF $F$ that weakly-satisfies (in expectation) $(c_0/(q+1))$-fraction of the bags of $\mathcal{I}(\mathbf{x}_0)$. Since all these bags bags are non-monochromatic, $F$ splits $(c_0/q)$-fraction of the bags of $\mathcal{I}(\mathbf{x}_0)$ (in expectation).

3. Observe that if $F$ splits $B \cup \{\mathbf{x}_0\}$ for a bag $B \in \mathcal{B}_1$, then either $F$ or $(1 - F)$ MIL-satisfies $B$. Thus, we output the best of $F$ or $(1 - F)$, one of which will satisfy (in expectation) $(c_0/(2(q+1)))$-fraction of the bags of $\mathcal{B}_1$ which is $(c_0/(4(q+1)))$-fraction of the bags of $\mathcal{B}$.

Combining all the above cases we complete the proof.

$\square$

## M   Generalization bounds for LLP-LTF$[q]$

The work of [37] had shown generalization bounds for LLP-LTF$[2]$ based on the bounds for the proportion function given [42]. The latter used the generalization bound result for multi-class labeling shown by [36] which applies to the generalization error w.r.t. any real-valued labeling function $f$ mapping the vector of labels $\mathbf{y}$ of a bag to $\mathbb{R}$ that is 1-Lipshitz w.r.t. the infinity norm over the domain of $\mathbf{y}$. We use the same approach below for LLP-LTF$[q]$ for any $q \geq 2$.

Firstly, it suffices to prove the result separately for each bag size $r \leq q$, since it can easily be shown that the bag-size composition is essentially preserved in the empirical training sample with high probability.

Let us fix a bag size $1 \leq r \leq q$. For each $t \in \{0, \ldots, r - 1\}$ define the labeling function $f_t : \{0,1\}^r \to \mathbb{R}$

$$f_t(\mathbf{y}) = \begin{cases} 0 & \text{if } \|\mathbf{y}\|_1 \leq t \\ 1 & \text{otherwise.} \end{cases} \tag{100}$$

Clearly, $|f(\mathbf{y}_1) - f(\mathbf{y}_2)| \leq |\mathbf{y}_1 - \mathbf{y}_2|_\infty$, and thus $f_t$ satisfies the condition above. Therefore, any function which correctly classifies bags of size $r$ into those with label proportion $< t/r$ vs. the rest admits generalization error bounds as given in [42, 36]. Thus, for any hypothesis which predicts the label proportion as $r$ different classes, we obtain $r - 1$ constraints using $t = 1, \ldots, r - 1$, on the per-class mis-classification errors. The final constraint is given by the fact that the fractions of bags of each of the $r$ classes sum to 1, thereby yielding the generalization bound on the sum of all the per-class mis-classification errors.