# OpenReview forum: "Algorithms and Hardness for Learning Linear Thresholds from Label Proportions"
_NeurIPS.cc/2022/Conference — NeurIPS 2022 Accept_

### Official Review · Reviewer_Bsrj · 2022-07-10

**Rating:** 7
**Confidence:** 3
**Soundness:** 4 excellent
**Presentation:** 4 excellent
**Contribution:** 3 good

**Summary:**

This paper studies the fairly natural learning problem of "learning from label proportions". The underlying problem is realizable supervised classification, however, instead of receiving labelled instances, the learner receives multiple bags of instances and the proportions of positively labelled instances in each bag, i.e., the "label proportions". This is motivated by e.g., privacy concerns. The goal of the learner is to find a hypothesis (from a hypothesis space) which predicts the instance labels in all bags in a way such that the number of bags with correctly predicted label proportions is maximised. This generalises regular supervised learning, as for bags of size 1 this corresponds to classifying all instance labels correctly. The main concern of the authors is to construct a polynomial-time "empirical risk minimiser" (or the equivalent thereof in this label proportions scenario) and not generalisation aspects (e.g., to new independently drawn bags).

In particular, the authors focus on this label proportions learning scenario in the case of linear threshold functions, i.e., halfspaces. For bags of size $\leq 2$ previous work suggested first algorithms and hardness results. This paper improves the hardness lower bound for bags of size at most 2 from essentially $1/2$ to essentially $4/9$. Additionally, this paper generalises these results to bags of size $\leq 3$ and achieves similar results there through an involved SDP-based relaxation. They also derive hardness lower bounds for the general case with bags of arbitrary size, which also applies to the improper case of functions that depend on a constant number of halfspaces, and propose an algorithm which asymptotically achieves this bound, however, using a weaker notion of satisfied bags.

They also prove a novel characterisation of psd matrices A,B satisfying $A\preccurlyeq B$, which might be of independent interest.

Finally, the theoretical results are complemented by first practical experiments, where the authors compare their approach with a simple randomised baseline on synthetic datasets.

**Questions:**

While the achieved results for linear thresholds are interesting, the question arises whether similar algorithms and bounds are possible in the general case (for arbitrary hypothesis space / set systems). Maybe there is some complexity notion similar to the VC dimension determining "learnability" here.

What about halfspaces with margin $\gamma$? Does this help in the learning scenario?

While it is true that for bag sizes $>2$ it is unclear whether $\langle r, x_1\rangle\langle r, x_2\rangle>0$ cannot be determined anymore, the label proportion still determines $\langle r, x_1\rangle\cdot\ \dots\ \cdot\langle r, x_q\rangle>0$. Can this be used by some algorithm? Please elaborate, as it seems like a natural generalisation of the $q=2$ case.

Do the authors thing the hardness bound of $4/9$ is best possible for $q=2$?

**Limitations:**

The paper only discusses the computational problem of finding a hypothesis such that a maximum number of bags is satisfied (have correctly predicted label proportions). However, as the problem is called "learning from label proportions" a discussion of possible generalisation bounds would be very interesting (prediction label proportions of unseen bags). Previous work, e.g., [28], apparently has generalisation bounds.

**Strengths And Weaknesses:**

Very well-written and easy to follow paper. The contributions will be most likely interesting to a theoretical sub-community of NeurIPS and potentially enables further work building on top.

The experiments are sufficient for such a theoretical contribution. In fact, they are not really necessary.

Minor points:
* Why do the authors not use the more common and simple term "halfspace" instead of LTF
* The two overview sections (1.4, 1.5) are helpful. To round it up the authors could include pointers to the actual Lemmas and Theorems (e.g., while discussing line 164 and following maybe note that the full statement is in Lemma 2.3, or multiple times on page 5).
* Please add a short section on "LLP-LTF[q]" with the most important details in the main paper, as well. For example, by moving the experimental evaluation to the appendix.
* (Standard) references for the dictatorship test, the lavel cover problem , and the folding trick would be very much appreciated.
* I would switch the order of "4. Experimental Evaluation" and "5. Hardness Result" to separate the theoretical results from the empirical ones more clearly.

Typos:
* Berry-Essen --> "Esseen"
* line 126: missing " " (space) between "proceeds.Since"

---

> ### Author Response · Authors · 2022-08-02
> **Author Response to Reviewer Bsrj**
>
> We are grateful to the reviewer for their encouraging feedback, helpful comments and suggestions.
>
> We answer the main questions and limitations mentioned by the reviewer below.
>
> Q. *While the achieved results for linear thresholds are interesting, the question arises whether similar algorithms and bounds are possible in the general case (for arbitrary hypothesis space / set systems). Maybe there is some complexity notion similar to the VC dimension determining "learnability" here.*
> *Authors*: Designing algorithms for LLP-learning more general (than LTF) classifiers is an interesting open problem. Our algorithms for LTFs can also be generalized to learning degree-$d$ polynomial threshold functions (PTFs) using degree-$d$ PTFs in the LLP setting (for constant $d$), because degree-$d$ PTFs can be represented as LTFs in a degree-$d$ tensor space.
> Regarding hardness bounds, under standard complexity assumptions (like P != NP) such bounds are likely only possible for restricted hypothesis classes (e.g. learning a concept class like LTFs using LTF or degree-d PTF hypothesis) unless some unexpected complexity consequences occur (see [Applebaum et al., *FOCS’08*]). VC-dimension or similar measures can also be used to restrict the concept and hypothesis classes.
> However, hardness bounds for unrestricted hypotheses *can* be shown using non-standard complexity  (such as hardness of random XOR) or cryptographic assumptions (see for e.g. [Daniely, *STOC'06*]). Both these approaches can be used to prove LLP complexity bounds.
>
> Q. *What about halfspaces with margin $\gamma$ ? Does this help in the learning scenario?*
> *Authors*: It is known (see for e.g. [Arriaga and Vempala, *Mach. Learn.’06*]) that random projection methods are able to efficiently learn halfspaces (and their intersections) with margin. Given constant $\gamma > 0$, a suitable random projection of $n$ points to an $O(\log n)$ dimensional space which will approximately preserve w.h.p all the normalized inner products among those $n$ points of magnitude at least $\gamma$, in particular their signs (See Corollary 2 of [Arriaga and Vempala, *Mach. Learn.’06*]). A subsequent brute force (yet poly$(n)$) search in the low-dimensional space will yield the desired halfspace.
> This approach also works for learning halfspaces in the LLP setting.
>
> Q. *While it is true that for bag sizes $ > 2$ it is unclear whether $\langle r, x_1\rangle\langle r, x_2\rangle > 0$ cannot be determined anymore, the label proportion still determines $\langle r, x_1\rangle\dots\langle r, x_q\rangle > 0$. Can this be used by some algorithm? Please elaborate, as it seems like a natural generalisation of the case.*
> *Authors*: It may be possible to use the Lasserre (i.e. sum-of-squares) SDP hierarchy to use cubic and higher products in some manner. However, we do not see any straightforward way to round such relaxations to obtain improved results.
>
> Q. *Do the authors thing the hardness bound of $4/9$ is best possible for $q=2$ ?*
> *Authors*: The techniques of this paper do not seem to yield any better than $4/9 + o(1)$ hardness for $q=2$, and we had not given much thought to improving the algorithm of [Saket, *NeurIPS’21*]. Either or both of these bounds can possibly be improved.
>
> Q. *A discussion of possible generalisation bounds would be very interesting (prediction label proportions of unseen bags).*
> *Authors*: Using the analysis of [28], [Saket, *NeurIPS’21*] includes a discussion (in Appendix E) of generalization bounds for $q=2$ and the strict bag satisfaction objective. For $q=3$, a similar bound would hold and we shall add a discussion of the same.
>
> We now address the minor points mentioned.
>
> - (For point 1) We use LTF for brevity and because it is more convenient to use in acronyms such as LLP-LTF.
>
> - (For points 2 and 4) We will include pointers to the actual lemmas and theorems, along with references to the dictatorship testing, label cover and folding.
>
> - (For point 3 and 5) We will do the suggested restructuring in the final version of the paper, using the additional page for adding content.
>
>
> We thank the reviewer for pointing out typos, which we shall correct along with any other typos in the paper.
>
> $ $
>
> **References**
>
> [Applebaum et al., *FOCS’08*] B. Applebaum, B. Barak, D. Xiao. On Basing Lower-Bounds for Learning on Worst-Case Assumptions. FOCS 2008. 211-220.
>
> [Arriaga and Vempala, *Mach. Learn.’06*]  	Rosa I. Arriaga, Santosh S. Vempala. An algorithmic theory of learning: Robust concepts and random projection. Mach. Learn. 63(2): 161-182. 2006.
>
> [Saket, *NeurIPS’21*] Reference [25] of the paper.
>
> [Daniely, *STOC’16*] 	Amit Daniely. Complexity theoretic limitations on learning halfspaces. STOC 2016.

---

> > ### Comment · Reviewer_Bsrj · 2022-08-03
> > **Thanks**
> >
> > Thanks for the clarifications!

---

### Official Review · Reviewer_gUha · 2022-07-12

**Rating:** 7
**Confidence:** 3
**Soundness:** 4 excellent
**Presentation:** 4 excellent
**Contribution:** 3 good

**Summary:**

The paper proposes algorithms for learning linear threshold functions (LTFs) from label proportions. In this model, the learning algorithm is given "bags" of points with the proportion of points in the bag labeled $1$. The goal is simply to find an LTF that maximizes the number of bags on which it labels the points exactly at the right proportion. The problem is NP-hard, so approximation algorithms are considered. The paper improves upon the previously known lower bound for bags of size $2$. It also gives a $\frac{1}{12}$ guarantee for bags of size $3$, and in general an $\Omega(1/q)$ guarantee for bags of size $q$. It is also shown that it's NP-hard to approximate the problem with bags of size $q$ beyond $\frac{1}{q}+o(1)$. The method is to solve a semidefinite programming relaxation, then round the result using a random hyperplane.

**Questions:**

What's the justification for choosing this objective function over, say, minimizing the total deviation or maximum deviation over all bags? Do your methods extend to other choices of objective? Are there previous results on other choices? Are these alternatives easier? harder?

**Limitations:**

None.

**Strengths And Weaknesses:**

The main contribution of the paper is the new SDP relaxation, which is new, non-trivial, and interesting.

The model has been studied before, and the bags input is justified by issues of privacy/legal constraints. However, I'm not completely convinced by the justification for the objective function (that for small bags it's reasonable). Why insist on getting as many bags as possible to have the exact input ratio, perhaps at the expense of gross errors on the other bags? Alternatively, one can try to minimize the total deviation, or the maximum deviation, or a host of other alternatives.

---

> ### Author Response · Authors · 2022-08-02
> **Author Response to Reviewer gUha**
>
> We thank the Reviewer for their encouraging feedback and helpful comments.
>
> We answer the questions asked by the reviewer below (cited references are added at the end of the response).
>
> Q. *What's the justification for choosing this objective function over, say, minimizing the total deviation or maximum deviation over all bags?*
> *Authors*: We feel that the (strict) objective of matching the bag label-proportion is a natural generalization of the “classification” objective in traditional supervised PAC learning in which a (feature-vector, binary-label) example is either classified correctly or incorrectly.  The strict bag objective is also a reasonable approximation to the deviation-minimization objective for small bags.
> More importantly, from a solution perspective, we feel that this strict bag objective allows for a compact and tractable SDP relaxation in which any feasible solution can be rounded to produce an LTF satisfying (in expectation) the approximation guarantee.
>
> Q. *Do your methods extend to other choices of objective?*
> *Authors*: For bags of sizes 2 and 3, it is possible (through a bit more analysis) to estimate the average deviation from the bag label proportions of the obtained LTF, but this performance degrades for larger bags. We do not see a simple way of enhancing our methods to provide better guarantees for the deviation-minimization objective, and will mention this as an interesting problem.
> We do, however, provide guarantees for the *weak satisfaction* objective (see Sec. 1.2 and Thm. 1.3 ) - a monochromatic bag is weakly satisfied if the LTF matches its label proportion, while a non-monochromatic bag is weakly satisfied if it remains non-monochromatic under the LTF’s labeling. This objective arises in situations when only limited label information is available per bag - whether it is all zeros, all ones, or otherwise. This is also similar to the fairly well studied *multiple instance learning* (MIL) setting in which we are only given the OR of the boolean labels of the feature-vectors of a bag (see [Bortsova at al., *MICCAI'18*], [Amores, *Artificial Intelligence'13*]). By guessing a 0-labeled feature-vector and adding it to each bag we can transform an MIL instance to a weakly satisfiable LLP instance. This allows us to apply our weak-satisfaction algorithm to provide guarantees for MIL learning LTFs, and we will add an explanation along with this summary.
>
> Q. *Are there previous results on other choices? Are these alternatives easier? Harder?*
> *Authors*: There are several LLP machine learning algorithms that attempt to fit ML models to a collection of  bags of feature-vectors and their label-proportions.  However, while being practically applicable, they do not provide any non-trivial worst case performance guarantees, even for learning LTFs in the LLP setting.
> Typically, the ML models have continuous (0,1)-outputs (predictions) on feature-vectors, and the ML algorithm attempts to minimize some loss between the label-proportion and the average prediction, summed over all the bags. The loss could be $\ell_1$, $\ell_2^2$, cross-entropy, KL-divergence, or more specialized variants. Examples of such methods can be found in [Yu et al., *ICML’13*], [Dulac-Arnold et al., *arXiv’19*], [Liu et al., *NeurIPS’19*], [Scott & Zhang, *NeurIPS’20*], and we will add the relevant references.
>
>
> $ $
>
>
> **References**
>
> [Bortsova at al., *MICCAI'18*]  Reference [6] in the paper.
>
> [Amores, *Artificial Intelligence'13*] J. Amores, “Multiple instance classification: Review, taxonomy and comparative study,” Artificial Intelligence, vol. 201, pp. 81–105, 2013.
>
> [Yu et al., *ICML’13*] Reference [27] in the paper.
>
> [Dulac-Arnold et al., *arXiv’19*] Reference [12] in the paper.
>
>  [Liu et al., *NeurIPS’19*] Jiabin Liu, Bo Wang, Zhiquan Qi, Yingjie Tian, Yong Shi. Learning from Label Proportions with Generative Adversarial Networks. NeurIPS 2019: 7167-7177.
>
> [Scott & Zhang, *NeurIPS’20*] Clayton Scott, Jianxin Zhang. Learning from Label Proportions: A Mutual Contamination Framework. NeurIPS 2020.

---

> > ### Comment · Reviewer_gUha · 2022-08-06
> > **Reaction**
> >
> > Thank you for the answers. I think it would be useful to give a similar (or briefer) discussion of the choice of objective in the introduction.

---

> > > ### Author Response · Authors · 2022-08-06
> > > **Author Reply**
> > >
> > > Thank you for your suggestion, which we accept. We will add a brief discussion to the introduction summarizing our response on the choice of the objective.

---

### Official Review · Reviewer_RxJW · 2022-07-18

**Rating:** 4
**Confidence:** 4
**Soundness:** 3 good
**Presentation:** 2 fair
**Contribution:** 2 fair

**Summary:**

This work aims to propose an algorithm and theoretical analysis for learning from the label proportion problem, where the labels are given in aggregated form as a proportion of true labels in a bag of features. The proposed algorithm is for the case when the size of the bags is less than three and it comes with guarantees on the fraction of satisfying bags. The theoretical analysis shows the hardness of the learning problem. Some experimental results on synthetic data are provided.

**Questions:**

- Can the authors discuss the complexity of the proposed algorithm?

**Limitations:**

Yes.

**Strengths And Weaknesses:**

I find the hardness results, saying that satisfying more than 1/q + O(1) fraction of bags in the learning from label proportion problem is NP-hard, to be interesting since most of the previous work focus on developing new algorithms while not  much looks into the inherent hardness of this problem. However, here are a few concerns:
- My main concern is the readability of this work. Even though the hardness result is interesting, I find it hard to understand the proofs. For the overview provided in Sec. 1.5, it would be helpful if the definitions of label cover problem and the template of a dictatorship test are formally stated to make this work more self-contained. Also, the proof in Sec. 5 is hard to parse since it is mostly formulas without intuitions. The readability of this proof needs to be improved, probably by making it more verbal. Besides, there are too many typos that harms readability a lot. See below.
- It seems that the complexity of the proposed algorithm is at least cubic in the size of features, which make it impractical. It would be helpful if the authors provide a detailed discussion on the complexity of the proposed algorithm. I wonder how the runtime increases as the number of dimension d increases.
- The empirical evaluation seems too toy and comparison with existing learning from label proportion algorithms are missing.
- Missing references on existing work on learning from label proportion:
Scott C, Zhang J. Learning from label proportions: A mutual contamination framework. Advances in neural information processing systems. 2020;33:22256-67.

Typos:
- Line 97: the lower case f is not defined.
- Line 97: it is unsplit by F is the latter ... -> it is unsplit by F if the latter ...
- Line 110: In the worst, case -> In the worst case,
- Line 129: r is not defined.
- At Line 150, the symbol (*) is used to denote a formula while later at Lemma 2.4 the same symbol (*) is used to denote another formula.
- Line 205: E is undefined.
- Line 310: A right parenthesis is missing.

---

> ### Author Response · Authors · 2022-08-02
> **Author Response to  Reviewer RxJW**
>
> We thank the reviewer for their feedback and helpful comments.
>
> We first address the reviewer’s concerns and questions.
>
> $ $
>
> Reviewer concerns about readability.
>
> Q. *For the overview provided in Sec. 1.5, it would be helpful if the definitions of label cover
> problem and the template of a dictatorship test are formally stated to make this work more self-contained.*
> *Authors*: Accepted. We will add references and informal definitions for label cover and dictatorship test.
>
> Q. *Also, the proof in Sec. 5 is hard to parse since it is mostly formulas without intuitions. The readability of this proof needs to be improved, probably by making it more verbal.*
> *Authors*: We respectfully point out that the proof in Section 5 is described in words on lines 213-216 of the overview in Sec. 1.5. We will, however, add in this part of the overview a pointer to Sec. 5 for ease of reference.
>
> $ $
>
> Reviewer concerns about the complexity of the algorithm.
>
> Q. *Can the authors discuss the complexity of the proposed algorithm?
> It seems that the complexity of the proposed algorithm is at least cubic in the size of features, which make it impractical. It would be helpful if the authors provide a detailed discussion on the complexity of the proposed algorithm. I wonder how the runtime increases as the number of dimension d increases.*
> *Authors*: Let us consider the SDP in Fig. 1 and Fig 2. It has the following $(d+1)\times(d+1)$ variable matrices which are constrained to be psd:
> - The matrix $R$
> - For each $3$-sized non-monochromatic bag $B =$ { ${\bf x}_i, {\bf x}_j, {\bf x}_k$}:
>   - the 3 matrices  $R^{(r,s)}$ for $r,s \in$ {$i,j,k$}, $r \neq s$.
>   - the 3 matrices $\overline{R}^{(r,s)} := R - R^{(r,s)}$ , for $r,s \in ${$i,j,k$}, $r \neq s$.
>   - the 3 matrices $\widehat{R}^{(r,s)}  := R^{(p,r)} + R^{(p,s)} - R$ , where {$p, r, s$}$ = ${$i,j,k$},  for $r,s \in ${$i,j,k$}, $r \neq s$.
>
>
> Let the total number of bags be $m$.  Construct a matrix $Z$ which has the above 9 matrices as block diagonal for each non-monochromatic bag, and an additional block diagonal for $R$. Constraining $Z \succeq 0$ ensures that all the block-diagonals are psd. The dimension of $Z$ is at most $(9m(d+1)+1)\times (9m(d+1)+1)$.
> There are $O(md^2)$ linear constraints to ensure the consistency of  $\overline{R}^{(r,s)}$ and $\widehat{R}^{(r,s)}$ defined above, and the equations (7), (8), (13), (14), (15) of the paper.
>
> We have a $N \times N$ psd matrix, where $N = O(md)$,  with $M = O(md^2)$  constraints.
>
> The SDP algorithm of [Jambulapati et al., *STOC’20*] takes $O(\sqrt{N}(M N^2 + N^\omega + M^\omega))$ time where $\omega$  is the matrix multiplication exponent. For us this gives an $O(m^{3.5}d^{4.5} + m^{\omega + 0.5}d^{2\omega + 0.5})$ running time.
> We think that our algorithm can be applied to small/medium scale problems using fast and approximate SDP solvers (e.g. [Aora et al., *FOCS’05*]). We will include the above discussion.
>
>
> $ $
>
>
> Reviewer concerns about empirical evaluation and missing reference.
>
> Q. *The empirical evaluation seems too toy and comparison with existing learning from label proportion algorithms are missing.*
> *Authors*: Our algorithm for satisfiable LLP-LTF$[3]$  provides worst case performance guarantees, unlike previous algorithms. We do not claim that our algorithm is state-of-the art for most real-life datasets. Our experimental evaluation is to demonstrate that our algorithm can be applied in certain situations to obtain high accuracy on bags as well as test feature-vectors. Comparison with the random LTF method shows that the high accuracy of our algorithm cannot be trivially obtained. This also holds true for the LLP-LTF$[4]$ experiments (Appendix I).
>
> We will add the reference to [Scott and Zhang, *NeurIPS’20*] which develops a new framework of LLP algorithms for various types of bag distributions.
>
> $ $
>
> We now address the typos:
>
> *Line 97: the lower case f is not defined* : It should be F instead of f, we will correct this.
> *Line 97: it is unsplit by F is the latter* ... -> *it is unsplit by F if the latter* …  : We will correct this.
> *Line 110: In the worst, case* -> *In the worst case,* : We will correct the comma placement.
>
>
> *Line 129: r is not defined*: We respectfully point out that on line 127 we explicitly state that the satisfying LTF is given by $\textnormal{pos}(\langle {\bf r},{\bf x}\rangle)$.  Nevertheless, we will add that ${\bf r}$ is the hyperplane normal vector.
>
> *At Line 150, the symbol* (\*) *is used to denote a formula while later at Lemma 2.4 the same symbol* (\*) *is used to denote another formula.* : We will change this to a different symbol to avoid any ambiguity.
>
> *Line 205: ${\sf E}$ is undefined* : We respectfully submit that ${\sf E}$ is the expectation operator which we feel is evident from its use. Nevertheless, we will add a sentence stating this.
>
> *Line 310: A right parenthesis is missing.* : We will add the parenthesis.

---

> ### Comment · Area_Chair_VQsm · 2022-08-08
> **Did the author response address your concerns?**
>
> Dear Reviewer RxJW,
>
> Did the author response address your concerns?
> If yes, then please acknowledge this in your review (or by responding to the author comments). If not, then please ask the authors a clarifying question during the author-reviewer discussion period (which lasts until this Tuesday Aug 9).
>
> Thanks!

---

### Meta-Review · Area_Chair_VQsm · 2022-08-27

**Recommendation:** Accept
**Confidence:** Certain

**Metareview:**

All of the reviewers found the theoretical results in this paper novel and significant. In particular, the main contribution of the paper, which is the new SDP relaxation, appears to be non-trivial and interesting. However, there remain concerns about readability of the paper, as outlined by one of the reviewers, and we request that the authors put some effort into addressing them.

**Award:**

No

---

### Decision · Program_Chairs · 2022-09-14

Accept